# FAITHFUL VISION-LANGUAGE INTERPRETATION VIA CONCEPT BOTTLENECK MODELS

**Songning Lai**[*,1,2,5]**, Lijie Hu**[†,*,1,2,4]**, Junxiao Wang**[1,2,4]**, Laure Berti-Equille**[3]**, and Di Wang**[†,1,2,4]

[1]Provable Responsible AI and Data Analytics (PRADA) Lab
[2]King Abdullah University of Science and Technology
[3]IRD - Institut de Recherche pour le Développement, Montpellier, France
[4]SDAIA-KAUST AI    [5]Shandong University

## ABSTRACT

The demand for transparency in healthcare and finance has led to interpretable machine learning (IML) models, notably the concept bottleneck models (CBMs), valued for their potential in performance and insights into deep neural networks. However, CBM's reliance on manually annotated data poses challenges. Label-free CBMs have emerged to address this, but they remain unstable, affecting their faithfulness as explanatory tools. To address this issue of inherent instability, we introduce a formal definition for an alternative concept called the Faithful Vision-Language Concept (FVLC) model. We present a methodology for constructing an FVLC that satisfies four critical properties. Our extensive experiments on four benchmark datasets using Label-free CBM model architectures demonstrate that our FVLC outperforms other baselines regarding stability against input and concept set perturbations. Our approach incurs minimal accuracy degradation compared to the vanilla CBM, making it a promising solution for reliable and faithful model interpretation.

> "The greatest obstacle to discovery is not ignorance; it is the illusion of knowledge."
>
> *- Daniel J. Boorstin*

## 1 INTRODUCTION

Contemporary machine learning models, like deep neural networks (DNNs), often rely on intricate nonlinear structures, making them challenging for end-users to understand and trust. This lack of interpretability hinders their adoption, especially in critical domains like healthcare (Ahmad et al., 2018; Yu et al., 2018; Lai et al., 2022) and finance (Cao, 2022), where transparency is crucial. In response to this demand, interpretable machine learning (IML) models (Doshi-Velez & Kim, 2017) offer explanations for their behavior and insights into their inner workings. Concept Bottleneck Models (CBMs) have emerged as a noteworthy IML model, focusing on enhancing system safety and building trust among stakeholders (Koh et al., 2020). Unlike traditional IML models, CBMs leverage elevated concepts, featuring a concept layer that aids in downstream tasks and produces easily understandable deep features, facilitating users' comprehension of the model's decision-making processes and concept relationships with input data.

CBMs have recently advanced significantly, driven by notable studies (Yuksekgonul et al., 2022; Oikarinen et al., 2023). These advancements are partly attributed to pre-trained large language models (LLMs) and vision-language models (VLMs) (Menon & Vondrick, 2023; Yang et al., 2024a;b; Xu et al., 2023b). Traditional CBMs require substantial manual annotation, which label-free CBM (Oikarinen et al., 2023) effectively addresses by leveraging factual information from pre-trained models, reducing the need for domain-specific experts. However, this convenience comes with inherent instability in pre-trained models, which we investigate in this paper. Instability is a common issue in deep learning interpretation methods, making it challenging to understand model reasoning (Hu et al., 2023c;b), especially with unlabeled data and self-supervised training (Ghorbani et al.,

---

*The first two authors contributed equally to this work.
†Correspondence to Lijie Hu {lijie.hu@kaust.edu.sa} and Di Wang {di.wang@kaust.edu.sa}.

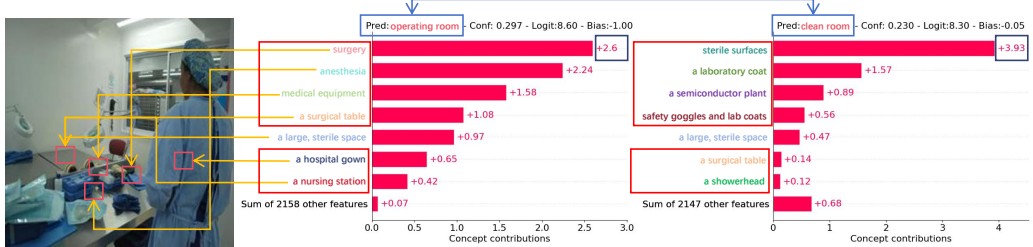

Figure 1: This is an example using the Places365 dataset (Zhou et al., 2017). The leftmost figure displays the input image, while the adjacent one on the left shows the concept output without perturbations. In contrast, the figure on the right presents the concept output after applying concept words and input perturbations, resulting in noticeable changes. These alterations include shifts in concept positioning along the ordinate and adjustments in the ranks of their respective weights, as exemplified by the concept "surgery". The prediction has also changed under slight perturbations.

2019; Dombrowski et al., 2019; Yeh et al., 2019). This instability issue also affects Label-free CBM, as Figure 1 illustrates.

To ensure a "faithful concept" for explaining deep features from the CBMs, we need a precise definition. Ideally, a "faithful concept" should have these four properties within the same concept set, which is generated by the LLM according to classes: (i) Significant overlap between the top-$k$ indices of the "faithful concept" and the original concept, ensuring interpretability. (ii) Inherent stability, with the concept vector remaining robust against random noise[1] and perturbations during LLM concept set generation. (iii) A prediction distribution close to that of the vanilla CBMs, preserving its outstanding performance. (iv) Stable output distribution, remaining robust during self-supervised learning and LLM concept set generation, even in the presence of perturbations.

**Our Contributions:** (1) *In-Depth Analysis of Faithfulness in VLMs:* We have identified a significant faithfulness issue in generating textual descriptions from visual data due to unstable alignment between text and images. (2) *Rigorous Definition of FVLC:* Building on our findings, we offer a formal definition of faithful vision-language interpretation with four essential properties, providing a clear framework for assessing VLM faithfulness. (3) *Innovative Faithful Mapping Module:* To tackle VLM faithfulness issues, we present an adaptable and flexible mapping module. It optimizes parameters minimally, preserving the integrity of the large model's pre-trained parameters, and aligns with our faithfulness definition. It smoothly integrates with classifier-free vision-language interpretation models without altering the original model structure, simplifying faithful interpretation in downstream tasks. (4) *Proposed Faithful Evaluation Metric:* Alongside the mapping module, we propose an original metric tailored for evaluating vision-language interpretation faithfulness, measuring alignment stability between textual information and visual content for more precise assessments. More details about the setting, implementation, and additional experimental results can be found in Appendix.

## 2 RELATED WORK

**Concept Bottleneck Models.** Considerable research has explored Concept Bottleneck Models (CBMs) recently (Havasi et al., 2022; Kim et al., 2023; Keser et al., 2023). However, CBMs face two primary challenges. Firstly, their performance often falls short compared to models without the concept bottleneck layer due to incomplete extraction of information from the original data into bottleneck features. Secondly, CBMs require laborious dataset annotation by human experts. Numerous studies have tackled these challenges and proposed potential solutions.

Chauhan et al. (2023) extend CBMs to interactive prediction settings by developing an interaction policy that determines which concepts to request labels for, thereby improving the final prediction. Post-hoc Concept Bottleneck models (PCBMs) (Yuksekgonul et al., 2022) can be applied to any neural network without sacrificing model performance while still providing interpretability benefits.

---

[1]It's important to highlight that the noise can originate not only from the input data but also from other sources like perturbed concept sets (e.g., Figure 1), emphasizing distinctions from adversarial robustness.

When concept annotations are unavailable, PCBMs demonstrate the ability to transfer concepts from other datasets or utilize natural language descriptions through multimodal models. Sawada & Nakamura (2022) integrate supervised concepts with unsupervised ones trained using self-explaining neural networks (SENNs) to obtain both types of concepts simultaneously. Oikarinen et al. (2023) discuss the limitations of existing CBMs and propose a Label-free CBM, which allows any neural network to be transformed into an interpretable CBM without the need for labeled concept data while maintaining high accuracy. Label-free CBM achieves state-of-the-art performance against other CBMs on classification tasks and represents an efficient unsupervised CBM. However, it is important to note that current CBM models do not consider the instability of concept sets' words generated by large language models or the instability caused by self-supervised training. Perturbations to the words in the concept set and input can impact the interpretability of the output. In this paper, our focus is primarily on enhancing the faithfulness and stability of Label-free CBMs.

**Faithfulness in explanation methods.** Faithfulness, as a crucial property of explanation models, entails accurately reflecting the true reasoning process of the underlying model in the explanation (Wiegreffe & Pinter, 2019; Herman, 2017; Jacovi & Goldberg, 2020; Lyu et al., 2022). It is closely related to other principles such as sensitivity, implementation invariance, input invariance, and completeness (Yeh et al., 2019). Completeness, for instance, signifies that an explanation should comprehensively encompass all relevant factors contributing to the prediction (Sundararajan et al., 2017). The remaining three principles pertain to stability concerning different types of perturbations. An explanation should undergo changes if significant features that influence the prediction are heavily perturbed (Adebayo et al., 2018) while remaining stable in the face of small perturbations (Yin et al., 2022). Therefore, stability plays a pivotal role in ensuring explanation faithfulness.

Several preliminary approaches have been proposed to achieve stable interpretations. For example, Yeh et al. (2019) theoretically analyzes the stability of post-hoc interpretations and proposes the use of smoothing techniques to improve interpretation stability. Additionally, Yin et al. (2022) design an iterative gradient descent algorithm to obtain counterfactual interpretations exhibiting desirable stability characteristics. However, these techniques are primarily designed for post-hoc interpretation and may not directly apply to CBMs. Novel methodologies need to be explored to address the stability requirements in CBMs. This paper aims to develop techniques that enhance stability and faithfulness within the context of CBMs. By considering the unique characteristics and challenges posed by CBMs, we seek to establish a framework that achieves stable interpretations while adhering to the principles of faithfulness, completeness, and sensitivity. Our proposed approach will facilitate reliable explanations in CBMs and contribute to the interpretable machine learning field.

## 3 PRELIMINARY

**Concept Bottleneck Models.** To introduce the original CBMs, we will adopt the notations used by Koh et al. (2020). We consider a classification task with a concept set denoted as $\mathcal{C} = \{p_1, \cdots, p_k\}$ and a training dataset represented as $\{(x_i, y_i, c_i)\}_{i=1}^N$, where for $i \in [N]$, $x_i \in \mathbb{R}^d$ is the feature vector, $y_i \in \mathbb{R}^{d_z}$ denotes the label, where $d_z$ corresponds to the number of classes, and $c_i \in \mathbb{R}^k$ denotes the concept vector whose $j$-th entry represents the weight of the concept $p_j$. In CBMs, we aim to learn two representations. One transforms from the input space to the concept space, which is represented by $g : \mathbb{R}^d \to \mathbb{R}^k$. The other one maps from the concept space to the prediction space, which can be denoted by $f : \mathbb{R}^k \to \mathbb{R}^{d_z}$. For any input $x$, we aim to make its predicted concept vector $\hat{c} = g(x)$ and prediction $\hat{y} = f(g(x))$ to be close to its underlying ones.

**Label-free CBMs.** The label-free CBM proposed by Oikarinen et al. (2023) has four steps:

Step 1: Concept set creation and filtering. In traditional CBMs, the concept set is typically generated through annotations by human experts. Instead, we propose utilizing the OpenAI API and leveraging GPT-3 (Brown et al., 2020) to automatically generate the concept set based on the classes, which are the dataset's labels. GPT-3 possesses a substantial amount of domain knowledge and can effectively identify important concepts associated with detecting each class when prompted appropriately. To accomplish this, we ask GPT-3 the following: (i) List the most important features for recognizing something as a {class}; (ii) List the things most commonly seen around a {class}; (iii) Give superclasses for the word {class}. After Step 1, we get a concept set $\mathcal{C} = \{p_1, \cdots, p_k\}$. See the Appendix C for additional details and sample output.

Step 2 and 3: Learning the Concept Bottleneck Layer (CBL). After obtaining the concept set, the following step involves acquiring a projection from the feature space of the backbone model into a space where interpretable concepts correspond to axis directions. To achieve this, we should learn the projection weights $W_c$ without relying on any labeled concept data by using CLIP-Dissect (Oikarinen & Weng, 2022; Guo et al., 2023b;a). Initially, we require a concept set that the bottleneck layer is supposed to represent as $\mathcal{C} = \{p_1, \cdots, p_k\}$, and a training dataset (such as images) without $c_i$ of the original task. We then calculate and save the CLIP concept activation matrix $M$, where $M_{i,j} = E_I(x_i) \cdot E_T(P_j)$, and $E_I$ and $E_T$ are the CLIP image and text encoders respectively. $W_c$ is initialized as a random $k \times d$ matrix, where $d$ is the dimensionality of backbone features $bf(\cdot)$ (Backbone is the part that extracts features from the input image for the network). The concept set $\mathcal{C}$ is created in Step 1, and the training dataset is provided by the downstream task. We define $g(\cdot) = W_c bf(\cdot)$, We use $l \in [d]$ to denote a neuron of interest in the projection layer and its activation pattern is denoted as $q_l$ where $q_l = [g_l(bf(x_1)), \cdots, g_l(bf(x_N))]$ with $q_l \in \mathbb{R}^N$ and $g_l(x)$ is the $l$-th coordinate of $g(x)$. Our optimization goal is to minimize the defined objective $L$ over $W_c$, which is as follows:

$$\mathcal{L}(W_c) = \sum_{i=1}^{k} -\text{sim}(P_i, q_i) = \sum_{i=1}^{k} -\frac{\bar{q_i}^3 \cdot \bar{M}_{:,i}^3}{||\bar{q_i}^3||_2 ||\bar{M}_{:,i}^3||_2}. \tag{1}$$

Here, $\bar{q}$ indicates vector $q$ normalized to have mean 0 and standard deviation 1.

Step 4: After successfully learning the Concept Bottleneck Layer, the next step involves training the final predictor using the fully connected layer $W_F \in \mathbb{R}^{d_z \times k}$, where $d_z$ represents the number of output classes. For each input $x$, we can get the class prediction distribution $y(x, \boldsymbol{c}) = W_F g(x)$. This process allows us to map the extracted concepts to the output classes, enabling the model to make accurate predictions based on the learned features.

# 4 FAITHFUL VISION-LANGUAGE CONCEPT

**Faithfulness issues in Label-free CBMs.** The Label-free CBM, as currently implemented using GPT-3, has inherent limitations that can impact its stability and reliability. One such limitation is the potential introduction of instability and perturbation to the concept sets due to the reliance on GPT-3 for concept generation. Additionally, it is challenging to prevent input images from being perturbed, which further compounds the issue. These limitations can result in unstable performance, undermining the trustworthiness and faithfulness of the model. While Label-free CBM shows promise in explaining model behavior without relying on labeled training datasets, it suffers from conceptual instability when faced with slight perturbations in the concept set and input data. This instability renders it unsuitable for real-world applications where robustness and reliability are crucial. By addressing these challenges, we can advance the field towards more trustworthy and robust deep learning models.

**What is a "faithful concept"?** A "faithful concept" is crucial for model interpretation. A "faithful concept" encapsulates a weight vector that exhibits stability and reliability, thereby instilling trust in its ability to offer accurate explanations of model behavior. When we refer to a "faithful concept", we mean a concept vector that remains stable even perturbations introduced to the input data or modifications applied to the concept set itself. These perturbations may arise due to various factors, such as noise or slight alterations in the wording of the concept set. However, a "faithful concept" should demonstrate resilience and maintain its explanatory power within an acceptable range despite these perturbations. By ensuring the stability and reliability of the concept, we can establish a foundation for faithful interpretable model explanations, thereby enhancing our understanding of complex deep learning systems. See Appendix A for details intuition of "faithful concept".

**Definition 1** (**Top-$k$ overlaps**). *This definition mainly measures the overlap of the first $k$ concept weights. We need to use the overlaps of top-k indices to measure the similarity on concept. For vector $x \in \mathbb{R}^d$, we define the set of top-k component $T_k(\cdot)$ as follow,*

$$T_k(x) = \{i : i \in [d] \text{ and } \{|\{x_j \geq x_i : j \in [d]\}| \leq k\}\}.$$

*And for two vectors $x$, $x'$, the top-k overlap function $V_k(x, x')$ is defined by the overlapping ratio between the top-k components of two vectors, i.e., $V_k(x, x') = \frac{1}{k}|T_k(x) \cap T_k(x')|$.*

Note that, in concept-based models, $\boldsymbol{c}$ could be seen as a function of input $x$, i.e., $\boldsymbol{c} = g(x)$. Thus, the faithful concept $\tilde{\boldsymbol{c}}$ can also be seen as a function of $x$, denoted as $\tilde{g}(x)$. Moreover, since our goal

is to replace the concept vector with a more robust one, we follow the previous model except for the procedure to produce the vector $\tilde{c}$. We define an FVLC as follows.

**Definition 2** (**Faithful Vision-Language Concept**). *Under the same concept space, i.e., under the set of concepts generated by GPT3 at one time, we call a matrix $\tilde{W}_c$ is a $(D, R, \alpha, \beta, k_1, k_2)$-Faithful Vision-Language Concept (FVLC) model for the vanilla concept if it satisfies for any input $x$:*

- *(Similarity of Explanation) $V_{k_1}(\tilde{g}(x), g(x)) \geq \beta_1$ for some $1 \geq \beta_1 \geq 0$;*

- *(Stability of Explanation) $V_{k_2}(\tilde{g}(x), \tilde{g}(x) + \rho) \geq \beta_2$ for some $1 \geq \beta_2 \geq 0$ and all $\|\rho\| \leq R_1$, where $\|\cdot\|$ is a norm and $R_1 \geq 0$;*

- *(Closeness of Prediction) $D(y(x, \tilde{c}), y(x, c)) \leq \alpha_1$ for some $\alpha_1 \geq 0$, where $D$ is some probability distance or divergence;*

- *(Stability of Prediction) $D(y(x, \tilde{c}), y(x, \tilde{c} + \delta)) \leq \alpha_2$ for all $\|\delta\| \leq R_2$, where $D$ is some probability distance or divergence, $\|\cdot\|$ is a norm and $R_2 \geq 0$,*

*where $\tilde{g}(x) = \tilde{W}_c bf(x)$, $y(x, c) = W_F g(x)$, and $y(x, \tilde{c}) = W_F \tilde{g}(x)$, $y(x, \tilde{c} + \delta) = W_F(\tilde{g}(x) + \delta)$. Moreover, for any given $x$, $\tilde{c} = \tilde{g}(x)$ is a $(D, R, \alpha, \beta, k_1, k_2)$-Faithful Vision-Language Concept (FVLC). Here, $\alpha = \min\{\alpha_1, \alpha_2\}$, $\beta = \max\{\beta_1, \beta_2\}$, and $R = \min\{R_1, R_2\}$.*

## 5 FVLC FRAMEWORK

**Sensitivity.** In the above definitions, two properties are considered: similarity and stability, relevant to both prediction and explanation. Notably, within the domain of explanation, our aim extends beyond ensuring explanation stability alone. We also strive for explanation sensitivity, a distinct criterion from that of prediction. The concept should demonstrate sensitivity when crucial features are excluded while maintaining stability when subjected to minor perturbations.

**The top-$k$ approach.** To achieve this, we employ the top-$k$ approach to preserve these crucial features, ensuring that the modified concept $\tilde{c}$ maintains similar explainability to the original concept. The top-$k$ approach involves two parameters, $k_1$ and $\beta_1$. $k_1$ can be considered as prior knowledge, indicating that we believe the top-$k_1$ indices of the concept play the most crucial role in making the prediction or that their corresponding features almost entirely determine the prediction. On the other hand, $\beta_1$ quantifies the extent to which $\tilde{c}$ inherits explainability from the original concept. When $\beta_1 = 1$, it signifies that the top-$k_1$ order of the entries in $\tilde{g}(x)$ remains the same as in the original concept. Therefore, it is desirable for $\beta_1$ to be close to 1.

**Stability.** The stability condition involves two parameters, $R_1$ and $\beta_2$, representing the robust region and level of stability, respectively. Ideally, if $\tilde{c}$ satisfies this condition with $R_1 = \infty$ and $\beta_2 = 1$, it indicates an extremely stable concept, resilient to any randomness or perturbations. In practice, we strive to have $R_1$ as large as possible and $\beta_2$ sufficiently close to 1.

**Prediction.** The last two conditions pertain to the similarity and stability of concept-based prediction. In the third condition, $\alpha_1$ quantifies the proximity between the prediction distribution based on $\tilde{c}$ and that based on the original concept. A value of $\alpha_1 = 0$ implies that $\tilde{c} = c$. Therefore, our goal is to minimize $\alpha_1$ as much as possible. Similarly, the stability condition involves two parameters, $R_2$ and $\alpha_2$, representing the robust region and level of stability, respectively. Ideally, if $\tilde{c}$ satisfies this condition with $R_2 = \infty$ and $\alpha_2 = 0$, it indicates an extremely stable concept, resistant to any randomness or perturbations. In practice, we aim for $R_2$ to be as large as possible and $\alpha_2$ to be sufficiently small. Based on these discussions, it is evident that Definition 2 provides a reasonable and consistent definition for ensuring the required properties of a faithful concept.

**Framework.** We have proposed a strict definition of FVLC. To build the FVLC, we freeze the image encoder and derive a minimum-maximum optimization problem with four conditions in the Definition 2. Specifically, the formula optimization problem targets the third condition (predicted proximity) and makes it subject to the other three conditions. So, by definitions, we can get a rough optimization problem and derive the following overall objective function:

$$\min_{\tilde{W}_c} \mathbb{E}_x[\lambda_1 D(y(x, \tilde{c}), y(x, c)) - \lambda_2 V_{k_1}(\tilde{g}(x), g(x)) + \lambda_3 \max_{||\delta|| \leq R_2} D(y(x, \tilde{c}), y(x, \tilde{c} + \delta))$$
$$- \lambda_4 \max_{||\rho|| \leq R_1} V_{k_2}(\tilde{g}(x), \tilde{g}(x) + \rho)],$$

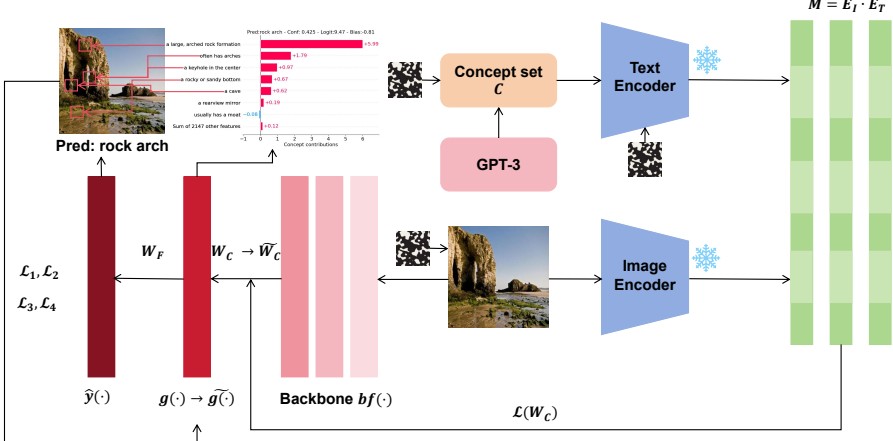

Figure 2: An overview of our pipeline for creating FVLC. For the concept set $\mathcal{C}$ output by GPT-3, it is input into the text encoder of CLIP to obtain $E_T$. The input image is processed by the image encoder of CLIP to obtain $E_I$. To accomplish the image classification task, the image is fed into the backbone to extract image features. The obtained activation matrix $M$ from $E_I$ and $E_T$ is used to learn the mappings $g(\cdot)$ and $W_c$ from the feature space to the concept space. Furthermore, the mapping $W_F$ from the concept space to the image category is learned. Then $\mathcal{L}_1/\mathcal{L}_2/\mathcal{L}_3/\mathcal{L}_4$ are employed to enhance the model's faithfulness ($W_c \rightarrow \tilde{W}_c$, $g(x) \rightarrow \tilde{g}(x)$). We introduce noise interference in the concept set, text encoder, and input image to validate the faithfulness of our model. The box mixed with black and white dots represents the noise, and snowflakes represent the freezing parameters.

where $\lambda_i, i \in [4]$ are hyperparameters. To solve this min-max optimization problem, we can generally use stochastic gradient descent-based methods to obtain the solution for the outer minimization and use PSGD (Projected Stochastic Gradient Descent) to solve the inner maximization. However, a major challenge is that the top-$k$ overlap functions $V_{k_1}(\tilde{g}(x), g(x))$ and $V_{k_2}(\tilde{g}(x), \tilde{g}(x) + \rho)$ are non-differentiable, which makes it difficult to use gradient descent. Therefore, we need to consider a surrogate loss for $-V_k(\cdot, \cdot)$. In Appendix B, we show details on such a surrogate loss and the optimization procedure. Finally, we have the following objective function:

$$\min_{\tilde{W}_c} \mathbb{E}_x[\lambda_1 \underbrace{D(y(x, \tilde{c}), y(x, c))}_{\mathcal{L}_1} + \lambda_2 \underbrace{\mathcal{L}_{k_1}(\tilde{g}(x), g(x))}_{\mathcal{L}_2} + \lambda_3 \underbrace{\max_{||\delta|| \leq R_2} D(y(x, \tilde{c}), y(x, \tilde{c} + \delta))}_{\mathcal{L}_3}$$

$$+ \lambda_4 \underbrace{\max_{||\rho|| \leq R_1} \mathcal{L}_{k_2}(\tilde{g}(x), \tilde{g}(x) + \rho)}_{\mathcal{L}_4}], \tag{2}$$

where $\mathcal{L}_k$ is the surrogate loss and $D$ is Kullback-Leibler Divergence.

## 6 EXPERIMENTS

### 6.1 EXPERIMENTAL SETUP

**Datasets.** We conducted a comprehensive evaluation of our approach by training our model on four diverse datasets: CIFAR-10, CIFAR-100 (Krizhevsky et al., 2009), CUB (Wah et al., 2011), and Places365 (Zhou et al., 2017). See Appendix D for more details, such as datasets, backbone, baselines, and other settings.

**Perturbations.** First, *word perturbation*. For the concept set generated by Step 1 in LCBM, we perform word perturbation. Taking advantage of the power of GPT-3 (Brown et al., 2020), we enter

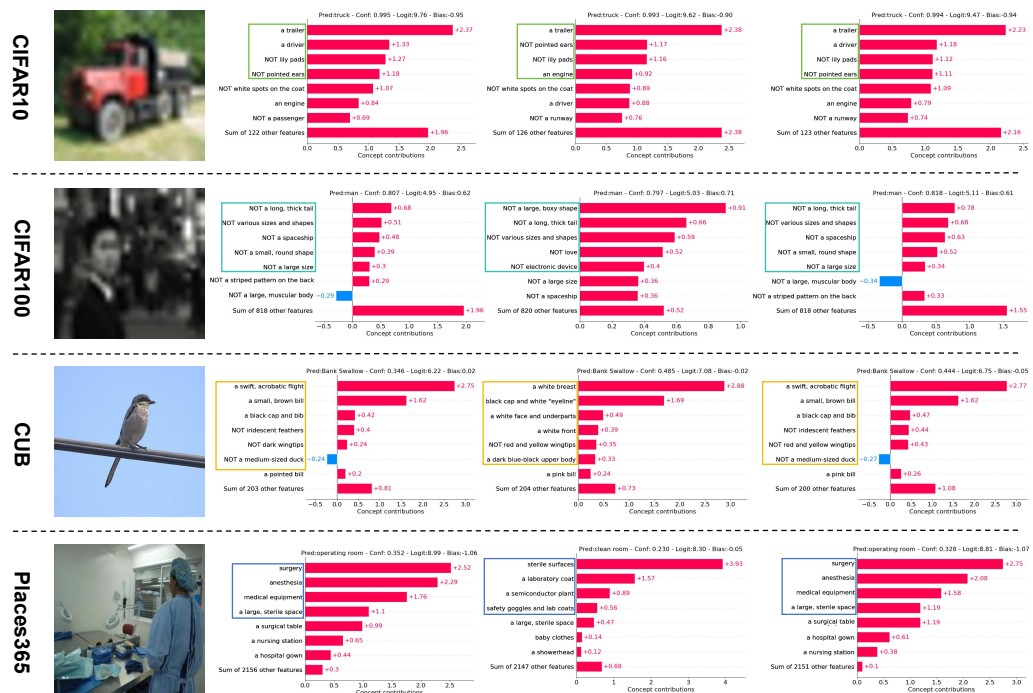

Figure 3: The visualizations for concept weights and final layer weights on one sample from each dataset. The sequence, from left to right, includes the input image $x$, concept weight visualization without perturbation ($c$), concept weight visualization with perturbation ($c + \delta$), and optimized concept weight visualization with perturbation ($\tilde{c} + \delta$). See Appendix E for a larger version of these subfigures.

| Method | CIFAR10 | CIFAR100 | CUB | Places365 |
|---|---|---|---|---|
| Standard (No interpretability) | 88.80% | 70.10% | 76.70% | 48.56% |
| P-CBM (CLIP) | 84.50% | 56.00% | N/A | N/A |
| Label-free CBM | 86.32% | 65.42% | 74.23% | 43.63% |
| WP1(5%) - base | 86.47% | 65.13% | 74.08% | 43.57% |
| WP1(5%) - **FVLC** | 86.34% | **65.43%** | 73.96% | 43.67% |
| WP1(10%) - base | 86.25% | 65.09% | 73.97% | 43.67% |
| WP1(10%) - **FVLC** | 86.39% | 64.90% | 73.92% | 43.62% |
| WP2 - base | 86.41% | 65.16% | 73.96% | 43.54% |
| WP2 - **FVLC** | 86.22% | 65.34% | **74.44%** | **44.55%** |
| IP - base | 86.62% | 65.36% | 74.39% | 43.64% |
| IP - **FVLC** | **86.88%** | 65.29% | 74.01% | 43.71% |
| WP1(5%)+WP2 - base | 86.49% | 65.17% | 73.90% | 43.67% |
| WP1(5%)+WP2 - **FVLC** | 86.43% | 65.33% | 73.92% | 43.49% |
| WP1(10%)+WP2 - base | 86.30% | 64.87% | 73.82% | 43.61% |
| WP1(10%)+WP2 - **FVLC** | 86.38% | 65.06% | 74.01% | 43.44% |
| WP1(10%)+WP2+IP - base | 85.96% | 64.41% | 73.74% | 43.32% |
| WP1(10%)+WP2+IP - **FVLC** | 86.70% | 65.14% | 74.36% | 43.46% |

Table 1: The table presents accuracy for baselines and FVLC before and after the perturbations across four benchmark datasets. In the first row, we have the standard backbone image classification model, which lacks interpretability. The latest CBM models with interpretability are P-CBM (Yuksekgonul et al., 2022) and Label-free CBM (Oikarinen et al., 2023), both of which employ an unsupervised process for generating concept sets, eliminating the need for manual labeling. The accuracy of Label-free CBM under various perturbations is displayed in while row color. The percentages in parentheses indicate the degree of added WP1 (The same as below).

the full concept set into GPT-3 and ask it to replace 5%, and 10% of words with synonyms. By this operation, we obtain the concept set with 5%, and 10% word perturbation. We name this perturbation **WP1** (Word Perturbation 1). Second, *word embedding perturbation*. Specifically, we choose the embedding $x$ of the last layer of the text encoding layer and then embed $x' = x + \mathcal{N}(0, \sigma)$ with a

perturbation of a certain radius $\sigma = 0.001$. We name this perturbation **WP2** (Word Perturbation 2). Third, *input perturbation*. We add the perturbation of a certain radius $\sigma = 0.001$ in input images (Input image pixels have been normalized). We name this perturbation **IP** (Input Perturbation).

| Method | CIFAR10 | | CIFAR100 | | CUB | | Places365 | |
|---|---|---|---|---|---|---|---|---|
| | TCPC | TOPC | TCPC | TOPC | TCPC | TOPC | TCPC | TOPC |
| WP1(5%) - base | 1.55E-01 | 6.32E-02 | 1.01E-01 | 7.17E-02 | 1.26E-01 | 1.85E-01 | 1.59E-01 | 6.40E-02 |
| WP1(5%) - **FVLC** | **1.12E-03** | **8.55E-03** | **2.81E-03** | **4.51E-03** | **1.05E-02** | **1.50E-03** | **1.38E-03** | **1.30E-03** |
| WP1(10%) - base | 1.99E-01 | 8.36E-02 | 1.94E-01 | 1.31E-01 | 2.32E-01 | 3.41E-01 | 2.26E-01 | 1.14E-01 |
| WP1(10%) - **FVLC** | **1.19E-03** | **7.40E-03** | **3.67E-03** | **4.55E-03** | **1.19E-02** | **1.53E-03** | **1.39E-03** | **1.25E-03** |
| WP2 - base | 1.53E-01 | 4.99E-02 | 1.36E-01 | 6.67E-02 | 1.43E-01 | 1.73E-01 | 1.40E-01 | 6.37E-02 |
| WP2 - **FVLC** | **1.10E-02** | **8.72E-03** | **3.35E-03** | **4.55E-03** | **1.05E-02** | **1.53E-03** | **1.55E-03** | **1.29E-03** |
| IP - base | 1.68E-01 | 6.28E-02 | 1.38E-01 | 8.81E-02 | 1.71E-01 | 2.23E-01 | 1.73E-01 | 8.09E-02 |
| IP - **FVLC** | **8.02E-03** | **8.29E-03** | **3.24E-03** | **4.56E-03** | **1.04E-02** | **1.59E-03** | **1.50E-03** | **1.25E-03** |
| WP1(5%)+WP2 - base | 1.85E-01 | 3.50E-02 | 1.28E-01 | 6.65E-02 | 1.44E-01 | 1.79E-01 | 1.60E-01 | 6.32E-02 |
| WP1(5%)+WP2 - **FVLC** | **1.20E-02** | **7.46E-03** | **3.67E-03** | **4.55E-03** | **9.81E-02** | **1.51E-03** | **1.54E-03** | **1.28E-03** |
| WP1(10%)+WP2 - base | 1.17E-01 | 8.62E-02 | 1.93E-01 | 1.32E-01 | 1.76E-01 | 3.45E-01 | 2.52E-01 | 1.17E-01 |
| WP1(10%)+WP2 - **FVLC** | **1.18E-02** | **9.41E-03** | **2.06E-02** | **1.44E-02** | **1.87E-02** | **3.79E-02** | **2.74E-02** | **1.18E-02** |
| WP1(10%)+WP2+IP - base | 1.36E-01 | 1.05E-01 | 2.22E-01 | 1.55E-01 | 1.95E-01 | 3.54E-01 | 2.62E-01 | 1.44E-01 |
| WP1(10%)+WP2+IP - **FVLC** | **1.43E-02** | **1.11E-02** | **2.39E-02** | **1.77E-02** | **2.21E-02** | **4.54E-02** | **3.35E-02** | **1.34E-02** |

Table 2: The table demonstrates the stability (TCPC and TOPC) results for both the baselines and FVLC approaches across four benchmark datasets subjected to various perturbations.

**Evaluation metrics.** In our experiments, we assessed the stability of our model using two metrics: Total Concept Perturbation Change (TCPC) and Total Output Perturbation Change (TOPC).

TCPC measures the stability of model interpretability between two concept weight vectors, namely $c_1$ (the concept weight vector before perturbation) and $c_2$ (the concept weight vector after perturbation). It is calculated as $\text{TCPC}(c_1, c_2) = \|c_1 - c_2\| / \|c_1\|$, where $c_1, c_2$ represents the complete concept weight vectors. TOPC assesses the stability of the model's output results by comparing two sets of outputs: $y_1$ (the output results before perturbation) and $y_2$ (the output results after perturbation). It is defined as $\text{TOPC}(y_1, y_2) = \|y_1 - y_2\| / \|y_1\|$, where $y_1, y_2$ represents the complete last layer output vector (i.e. the classes prediction output vector).

## 6.2 UTILITY EVALUATION

Table 1 presents the accuracy results of our proposed FVLC method and the baseline approach on four datasets with different levels of perturbations. The table clearly demonstrates that our method maintains a consistent and high accuracy across all datasets without any significant changes or losses. This highlights the robustness of our approach in terms of accuracy preservation. Moreover, our FVLC method outperforms the baseline approach in terms of accuracy in some cases. This indicates that our approach not only maintains stability but also achieves improved accuracy compared to the baseline method. Overall, the results presented in Table 1 affirm the effectiveness of our FVLC method, which successfully combines high accuracy with interpretability while preserving stability across various datasets.

## 6.3 STABILITY EVALUATION

Table 2 analyzes the TCPC and TOPC values, which measure the stability of the CBMs under various perturbations. The table demonstrates that, in comparison to the baseline approach, FVLC exhibits higher stability in terms of concept weights and smaller differences in the output result matrix before and after perturbation. These findings indicate that FVLC combines interpretability with strong perturbation resistance, making it a faithful model. The experimental results reveal that, on average, FVLC outperforms the baseline model in terms of both TCPC and TOPC, with an average reduction of 90%.

For brevity, Figure 3 presents visualizations of the concept set differences and concept weight changes, both with and without fine-tuning, before and after perturbation for each dataset. These visualizations provide additional insights into the effectiveness and stability of FVLC in handling perturbations. The results presented in Table 2 and Figure 3 collectively demonstrate the superior stability of FVLC compared to the baseline model. FVLC successfully achieves faithful interpretation, making it a promising approach for concept-based modeling. We also carried out repeated experiments in multiple concept spaces to further verify that our method conforms to Definition 2. More experiments are in the Appendix F.

| Method | Setting | | | CIFAR10 | | CIFAR100 | | CUB | | Places365 | |
|---|---|---|---|---|---|---|---|---|---|---|---|
| | $\mathcal{L}_2$ | $\mathcal{L}_3$ | $\mathcal{L}_4$ | TCPC | TOPC | TCPC | TOPC | TCPC | TOPC | TCPC | TOPC |
| **WP1(10%) - FVLC** | | | | 1.99E-01 | 8.36E-02 | 1.94E-01 | 1.31E-01 | 2.32E-01 | 3.41E-01 | 2.26E-01 | 1.14E-01 |
| | ✓ | | | 2.09E-02 | 3.14E-02 | 2.81E-02 | 4.88E-02 | 4.08E-01 | 7.56E-02 | 4.69E-02 | 6.14E-02 |
| | | ✓ | | 1.80E-02 | 1.79E-02 | 2.01E-02 | 2.85E-02 | 3.77E-01 | 4.50E-02 | 4.48E-02 | 3.68E-02 |
| | | | ✓ | 4.78E-03 | 3.11E-02 | 1.67E-02 | 2.19E-02 | 5.69E-02 | 5.52E-03 | 6.04E-03 | 4.98E-03 |
| | ✓ | ✓ | | 1.67E-02 | 1.11E-02 | 5.39E-02 | 6.85E-02 | 1.69E-01 | 2.12E-02 | 1.98E-02 | 1.81E-02 |
| | ✓ | | ✓ | 1.65E-03 | 1.01E-02 | 5.08E-03 | 6.43E-03 | 1.61E-02 | 2.14E-03 | 1.95E-03 | 1.72E-03 |
| | | ✓ | ✓ | 1.63E-03 | 1.02E-02 | 5.04E-03 | 6.27E-03 | 1.63E-02 | 2.10E-03 | 1.94E-03 | 1.71E-03 |
| | ✓ | ✓ | ✓ | **1.19E-03** | **7.40E-03** | **3.67E-03** | **4.55E-03** | **1.19E-02** | **1.53E-03** | **1.39E-03** | **1.25E-03** |
| **WP2 - FVLC** | | | | 1.53E-01 | 4.99E-02 | 1.36E-01 | 6.67E-02 | 1.43E-01 | 1.73E-01 | 1.40E-01 | 6.37E-02 |
| | ✓ | | | 7.62E-02 | 2.29E-02 | 2.02E-02 | 2.08E-02 | 6.09E-02 | 5.22E-03 | 1.04E-02 | 5.31E-03 |
| | | ✓ | | 5.37E-02 | 1.84E-02 | 1.46E-02 | 1.16E-02 | 6.80E-02 | 2.95E-03 | 1.06E-02 | 2.36E-03 |
| | | | ✓ | 5.19E-02 | 1.46E-02 | 1.31E-02 | 9.34E-03 | 5.30E-02 | 3.24E-03 | 6.83E-03 | 2.12E-03 |
| | ✓ | ✓ | | 4.57E-02 | 1.82E-02 | 1.45E-02 | 1.01E-02 | 5.01E-02 | 3.46E-03 | 6.30E-03 | 2.94E-03 |
| | ✓ | | ✓ | 2.73E-02 | 6.96E-03 | 7.10E-03 | 4.60E-03 | 2.42E-02 | 1.91E-03 | 3.97E-03 | 2.36E-03 |
| | | ✓ | ✓ | 2.47E-02 | 8.00E-03 | 5.29E-03 | 5.25E-03 | 2.40E-02 | 1.70E-03 | 3.82E-03 | 1.96E-03 |
| | ✓ | ✓ | ✓ | **1.10E-02** | **8.72E-03** | **3.35E-03** | **4.55E-03** | **1.05E-02** | **1.53E-03** | **1.55E-03** | **1.29E-03** |
| **IP - FVLC** | | | | 1.68E-01 | 6.28E-02 | 1.38E-01 | 8.81E-02 | 1.71E-01 | 2.23E-01 | 1.73E-01 | 8.09E-02 |
| | ✓ | | | 6.39E-02 | 2.55E-02 | 2.56E-02 | 3.57E-02 | 8.04E-02 | 1.27E-02 | 1.16E-02 | 9.80E-03 |
| | | ✓ | | 3.82E-02 | 3.91E-02 | 1.53E-02 | 2.16E-02 | 4.77E-02 | 7.66E-03 | 6.99E-03 | 5.88E-03 |
| | | | ✓ | 1.63E-02 | 1.73E-02 | 6.35E-03 | 9.35E-03 | 2.14E-02 | 3.30E-03 | 3.01E-03 | 2.59E-03 |
| | ✓ | ✓ | | 2.04E-02 | 2.10E-02 | 8.35E-03 | 1.15E-02 | 2.67E-02 | 4.04E-03 | 3.77E-03 | 3.18E-03 |
| | ✓ | | ✓ | 1.23E-02 | 1.27E-02 | 5.09E-03 | 6.94E-03 | 1.65E-02 | 2.45E-03 | 2.35E-03 | 1.94E-03 |
| | | ✓ | ✓ | 1.26E-02 | 1.09E-02 | 5.07E-03 | 7.00E-03 | 1.59E-02 | 2.53E-03 | 2.29E-03 | 1.95E-03 |
| | ✓ | ✓ | ✓ | **8.02E-03** | **8.29E-03** | **3.24E-03** | **4.56E-03** | **1.04E-02** | **1.59E-03** | **1.50E-03** | **1.25E-03** |
| **WP1(10%)+WP2 - FVLC** | | | | 1.17E-01 | 8.62E-02 | 1.93E-01 | 1.32E-01 | 1.76E-01 | 3.45E-01 | 2.52E-01 | 1.17E-01 |
| | ✓ | | | 5.97E-02 | 6.28E-02 | 1.43E-01 | 9.94E-02 | 1.24E-01 | 2.58E-01 | 1.86E-01 | 8.24E-02 |
| | | ✓ | | 3.76E-02 | 6.34E-02 | 1.48E-01 | 1.03E-01 | 1.21E-01 | 2.63E-01 | 1.90E-01 | 8.13E-02 |
| | | | ✓ | 3.52E-02 | 2.54E-02 | 6.03E-02 | 4.13E-02 | 5.08E-02 | 1.07E-01 | 7.63E-02 | 3.31E-02 |
| | ✓ | ✓ | | 9.21E-02 | 5.45E-02 | 1.28E-01 | 8.56E-02 | 1.04E-01 | 2.24E-01 | 1.60E-01 | 6.82E-02 |
| | ✓ | | ✓ | 3.15E-02 | 2.90E-02 | 6.88E-02 | 4.71E-02 | 5.76E-02 | 1.21E-01 | 8.86E-02 | 3.72E-02 |
| | | ✓ | ✓ | 1.53E-02 | 1.18E-02 | 2.78E-02 | 1.89E-02 | 2.37E-02 | 4.88E-02 | 3.62E-02 | 1.55E-02 |
| | ✓ | ✓ | ✓ | **1.18E-02** | **9.41E-03** | **2.06E-02** | **1.44E-02** | **1.87E-02** | **3.79E-02** | **2.74E-02** | **1.18E-02** |
| **WP1(10%)+WP2+IP - FVLC** | | | | 1.36E-01 | 1.05E-01 | 2.22E-01 | 1.55E-01 | 1.95E-01 | 3.54E-01 | 2.62E-01 | 1.44E-01 |
| | ✓ | | | 7.37E-02 | 5.75E-02 | 1.29E-01 | 9.24E-02 | 1.21E-01 | 2.34E-01 | 1.74E-01 | 7.30E-02 |
| | | ✓ | | 7.85E-02 | 5.85E-02 | 1.23E-01 | 9.17E-02 | 1.14E-01 | 2.43E-01 | 1.70E-01 | 6.80E-02 |
| | | | ✓ | 4.59E-02 | 3.67E-02 | 7.62E-02 | 5.57E-02 | 6.74E-02 | 1.41E-01 | 1.08E-01 | 4.23E-02 |
| | ✓ | ✓ | | 5.81E-02 | 4.52E-02 | 9.69E-02 | 7.11E-02 | 8.99E-02 | 1.83E-01 | 1.34E-01 | 5.49E-02 |
| | ✓ | | ✓ | 5.84E-02 | 4.53E-02 | 9.75E-02 | 7.17E-02 | 8.94E-02 | 1.85E-01 | 1.36E-01 | 5.41E-02 |
| | | ✓ | ✓ | 2.91E-02 | 2.31E-02 | 4.85E-02 | 3.56E-02 | 4.46E-02 | 9.33E-02 | 6.78E-02 | 2.81E-02 |
| | ✓ | ✓ | ✓ | **1.43E-02** | **1.11E-02** | **2.39E-02** | **1.77E-02** | **2.21E-02** | **4.54E-02** | **3.35E-02** | **1.34E-02** |

Table 3: The ablation study of FVLC. We assess the efficacy of $\mathcal{L}_2$, $\mathcal{L}_3$, and $\mathcal{L}_4$ in Equation (2) when applied alongside perturbations to both the word and input.

## 6.4 ABLATION STUDY

In the ablation study, we conducted a thorough evaluation of each module (regularization) outlined in Equation (2). Our evaluation aimed to assess the significance and effectiveness of each module in enhancing the performance of our model. To accomplish this, we initially designated $\mathcal{L}_1$ as the primary loss function, which is defined as the first term in Equation (2). Subsequently, we systematically examined various combinations by selectively removing $\mathcal{L}_2$, $\mathcal{L}_3$, and $\mathcal{L}_4$. The outcomes of our study, presented in Table 3, unequivocally demonstrate that each regularizer incorporated in our objective function is both indispensable and effective. Each module contributes uniquely to the overall performance improvement of the model. Of particular significance is the inclusion of $\mathcal{L}_4$, which significantly enhances the model's stability. By ensuring the preservation of important concepts even under certain perturbations, the addition of $\mathcal{L}_4$ guarantees a higher level of robustness and stability in the model's predictions. This finding showcases the critical role played by this regularization module in refining the model's performance.

## 7 CONCLUSION

In this paper, we present a comprehensive definition of the Faithful Vision-Language Concept (FVLC), which establishes a robust and accurate interpretation of deep learning predictions. Our definition encompasses four key properties: similarity of explanation, stability of explanation, closeness of prediction, and stability of prediction. To obtain FVLC, we propose a novel method and evaluate it extensively through comprehensive experiments. Experimental results demonstrate that FVLC achieves comparable accuracy to the baseline while exhibiting stronger concept stability, indicating that FVLC is a more faithful explanation tool.

## 8 Acknowledgements

Di Wang, Junxiao Wang, Songning Lai, and Lijie Hu are supported in part by the baseline funding BAS/1/1689-01-01, funding from the CRG grand URF/1/4663-01-01, FCC/1/1976-49-01 from CBRC, and funding from the AI Initiative REI/1/4811-10-01 of King Abdullah University of Science and Technology (KAUST). Di Wang, Junxiao Wang, and Lijie Hu are also supported by the funding of the SDAIA-KAUST Center of Excellence in Data Science and Artificial Intelligence (SDAIA-KAUST AI).

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

## A   MORE DISCUSSION ON FAITHFUL CONCEPT

**What is a "faithful concept"?** As we can see from the previous discussion since the key component of Label-free CBM is the concept bottleneck layer, to get above faithful Label-free CBM, it is sufficient for us to get a faithful concept bottleneck layer, which is called "faithful concept". Before diving into our rigorous definition of "faithful concept", we first need to intuitively think about what properties it should have.

The first one is keeping an interpretability similar to the vanilla CBL. In the vanilla CBL for an input image, we can easily see that the rank of entries in the output of CBL, i.e., concept vector, can reflect the importance of each concept in the concept set. Thus, the "faithful concept" should also have almost the same order for each entry as in the vanilla one. However, keeping the rank for all entries is too stringent, motivated by the fact that the interpretability and the prediction always rely

on the most important entries. Here, we can relax the requirement to keep the top-k indices almost unchanged.

In fact, such a property is not enough as its interpretability could also be unstable to different perturbations. Modeling such instability is challenging as the perturbation could be caused by multi-resources, which is significantly different from the adversarial robustness. Our key observation is that wherever a perturbation comes from, it will subsequently change the output of CBL. Thus, if the interpretability of "faithful concept", i.e., the top-k indices, is resilient to noise, we can naturally think it is robust to those different perturbations.

However, even with the previous two properties, it is still insufficient. The main reason is that keeping interpretability does not indicate keeping the prediction performance. This is because keeping interpretability can only guarantee the rank of indices unchanged, but cannot ensure the magnitude of these entries, which determine the prediction unchanged. For example, suppose the vanilla concept vector is $(0.5, 0.3, 0.2)$, then the above "faithful concept" vector might be $(0.9, 0.051, 0.049)$, which is significantly different from the original one. Based on these, we should also enforce the prediction performance, i.e., the output distribution, to be almost the same as the vanilla one. Moreover, its output distribution should also be robust to perturbations motivated by Figure 1.

Based on our above discussion, our takeaway is that the "faithful concept" vector should make its top-$k$ indices and output distribution almost the same as the vanilla concept vector while also being robust to perturbations. Based on the above intuitions, we can translate the previous intuitions into rigorous mathematical definitions. Specifically, we call the above "faithful concept" as Faithful Vision-Language Concept (FVLC).

**Differences with adversarial robustness.** While both FVLC and adversarial robustness consider perturbations or noises on input data. There are many critical differences: (1) Not only the prediction, we should also make the interpretability stable and close to the vanilla concept vector. Due to these additional conditions, our method for achieving FVLC is totally different from the methods in adversarial robustness, such as certified robustness or adversarial training. See Section B for details. (2) The way of modeling robustness in FVLC is also totally different from adversarial robustness. In adversarial robustness, it usually models the robustness to perturbation on input data. However, in FVLC, due to the requirement on interpretability, i.e., the top-$k$ indices of the vector, we cannot adopt the same idea. Firstly, directly requiring the top-$k$ indices robust to perturbation on the input will make the optimization procedure challenging (which is a minimax optimization problem) as the top-$k$ indices function is non-differentiable, and calculating the gradient of the backbone neural network is costly. Secondly, rather than perturbation on input data, as we mentioned, the perturbation could come from multi-resources, such as embedding perturbation or a combination of perturbations. Thus, from this perspective, our stability of interpretability is more suitable for "faithful concept".

## B    OPTIMIZATION FOR FVLC LAYER

In the last section, we presented a rigorous definition of FVLC. To find such an FVLC, we propose to formulate a min-max optimization problem that involves the four conditions in Definition 2. Specifically, the formulated optimization problem takes the third condition (closeness of prediction) as the objective and subjects it to the other three conditions. Thus, we can get a rough optimization problem according to the definition. Specifically, we first have

$$\min_{\tilde{W}_c} \mathbb{E}_x D(y(x, \tilde{\boldsymbol{c}}), y(x, \boldsymbol{c})) \tag{3}$$

Equation (3) is the basic optimization goal, that is, we want to get a vector which has similar output prediction with the vanilla CBM for all input $x$. If there is no further constraint, then we can see the minimizer of (3) is just the vanilla CBM matrix $W_c$ that optimizes 1. We then consider constraints for this objective function:

| Notation | Remark | Notation | Remark |
|---|---|---|---|
| $\mathcal{C}$ | concept set | $x$ | an input data |
| $c_i$ | concept vector | $g : \mathbb{R}^d \to \mathbb{R}^k$ | representation which transforms from the input space to the concept space |
| $f : \mathbb{R}^k \to \mathbb{R}^{d_z}$ | representation which maps from the concept space to the prediction space | $\hat{c}$ | $g(x)$ |
| $\hat{y}$ | $f(g(x))$ | $M$ | concept activation matrix |
| $E_I$ | image encoder | $E_T$ | text encoder |
| $bf(\cdot)$ | backbone features extractor | $W_c$ | original concept embedding matrix |
| $W_F$ | weight matrix of the output layer | $\tilde{W}_c$ | FVLC |
| $V_k$ | top-$k$ overlap function | $\mathcal{L}_k$ | a surrogate loss of $-V_k$ |
| $D$ | divergence metric | $\tilde{g}(x)$ | $\tilde{W}_c bf(x)$ |
| $\tilde{c}$ | $\tilde{g}(x)$ | $y(x,c)$ | $W_F g(x)$ |
| $y(x,\tilde{c})$ | $W_F \tilde{g}(x)$ | $y(x,\tilde{c}+\delta)$ | $W_F(\tilde{g}(x)+\delta)$ |
| $\lambda_1, \lambda_2, \lambda_3, \lambda_4$ | regularization parameters | $R, \alpha, \beta, k1, k2$ | parameters in FVLC |

Table 4: Table of notations.

$$\forall x \text{ s.t. } \max_{||\delta|| \leq R_2} D(y(x,\tilde{c}), y(x,\tilde{c}+\delta)) \leq \alpha_2, \tag{4}$$

$$V_{k_1}(\tilde{g}(x), g(x)) \geq \beta_1, \tag{5}$$

$$\max_{||\rho|| \leq R_1} V_{k_2}(\tilde{g}(x), \tilde{g}(x)+\rho) \geq \beta_2; \tag{6}$$

where $\tilde{g}(x) = \tilde{W}_c bf(x)$, and $y(x,\tilde{c}) = W_F \tilde{g}(x)$, $y(x,\tilde{c}+\delta) = W_F(\tilde{g}(x)+\delta)$.

Equation (4) is the constraint of stability, Equation (5) corresponds to the condition of similarity of explanation, and Equation (6) links to the stability of explanation. Combining equations (3)-(6) and using regularization to deal with constraints, we can get the following min-max stochastic optimization problem.

$$\min_{\tilde{W}_c} \mathbb{E}_x[D(y(x,\tilde{c}), y(x,c)) - \lambda_1 V_{k_1}(\tilde{g}(x), g(x)) + \lambda_2 \max_{||\delta|| \leq R_2} D(y(x,\tilde{c}), y(x,\tilde{c}+\delta))$$
$$- \lambda_3 \max_{||\rho|| \leq R_1} V_{k_2}(\tilde{g}(x), \tilde{g}(x)+\rho)],$$

where $\lambda_1 > 0$, $\lambda_2 > 0$, and $\lambda_3 > 0$ are hyperparameters.

To solve this min-max optimization problem, we can generally use stochastic gradient descent-based methods to obtain the solution for the outer minimization and use PSGD (Projected Stochastic Gradient Descent) to solve the inner maximization. However, a major challenge is that the top-$k$ overlap functions $V_{k_1}(\tilde{g}(x), g(x))$ and $V_{k_2}(\tilde{g}(x), \tilde{g}(x)+\rho)$ are non-differentiable, which makes it impossible to use gradient descent. Therefore, we need to consider a surrogate loss for $-V_k(\cdot)$.

Our goal is to design a surrogate loss function $\mathcal{L}_k(\cdot)$ for $-V_k(\cdot)$ that can be used in training. We focus on the example of $\mathcal{L}_k(\tilde{c})$ for $-V_k(\tilde{c}, c)$. One possible naive surrogate objective is to use a distance metric, such as the $\ell_1$-norm, between $\tilde{c}$ and $c$, i.e., $L(\tilde{c}) = ||\tilde{c} - c||_1$. While this objective can preserve the top-$k$ overlap when we obtain the optimal or near-optimal solution, it lacks consideration of the top-$k$ information, making it a loose surrogate loss.

To address this issue, we propose minimizing the distance between $\tilde{c}$ and $c$ constrained on the top-$k$ entries only. Specifically, we minimize $||c_{S_c^k} - \tilde{c}_{S_c^k}||_1$, where $c_{S_c^k}, \tilde{c}_{S_c^k} \in \mathbb{R}^k$ are the vectors $c$ and $\tilde{c}$, respectively, constrained on the top-$k$ indices set $S_c^k$ of $c$. We use both top-$k$ indices sets for both vectors to involve the top-$k$ indices formation. Therefore, our surrogate loss function is:

$$\mathcal{L}_k(c, \tilde{c}) = \frac{1}{2k}(||c_{S_c^k} - \tilde{c}_{S_c^k}||_1 + ||\tilde{c}_{S_{\tilde{c}}^k} - c_{S_{\tilde{c}}^k}||_1). \tag{7}$$

Note that besides the $\ell_1$-norm, we can use other norms. However, in practice, we find that $\ell_1$-norm achieves the best performance. Thus, throughout the paper, we only use $\ell_1$-norm. By using this surrogate function, we have our relaxed function:

$$\min_{\tilde{W}_c} \mathbb{E}_x[D(y(x,\tilde{c}), y(x,c)) + \lambda_1 \underbrace{\mathcal{L}_{k_1}(\tilde{g}(x), g(x))}_{\mathcal{L}_2} + \lambda_2 \underbrace{\max_{||\delta|| \le R_2} D(y(x,\tilde{c}), y(x,\tilde{c}+\delta))}_{\mathcal{L}_3}$$

$$+ \lambda_3 \underbrace{\max_{||\rho|| \le R_1} \mathcal{L}_{k_2}(\tilde{g}(x), \tilde{g}(x)+\rho)}_{\mathcal{L}_4}].$$

Inspired by PGD in Madry et al. (2018), to solve the above optimization problem, in each iteration, with fixed current model $\tilde{W}_c^{t-1}$ (thus fixed $\tilde{g}(x)$) we have to first update $\delta$ and $\rho$. Specifically, in the $p$-th iteration for updating current noise $\delta_{p-1}^*$ we have:

$$\delta_p = \delta_{p-1}^* + \frac{\gamma_p}{|A_{p-1}|} \sum_{x \in A_{p-1}} \nabla_\delta D(y(x,\tilde{c}), y(x, \tilde{c}+\delta_{p-1}^*));$$

$$\delta_p^* = \arg\min_{||\delta|| \le R} ||\delta - \delta_p||,$$

where $A_{p-1}$ is a batch and $\gamma_p$ is a step size parameter for PGD.

Similarly, we can use the PGD and the surrogate loss of $\mathcal{L}_k(\cdot)$ to get the optimal $\rho^*$ in the $t$-th iteration of outer SGD.

$$\rho_q = \rho_{q-1}^* + \frac{\tau_q}{|B_{q-1}|} \sum_{x \in B_{p-1}} \nabla_\rho \mathcal{L}_{k_2}(\tilde{c}, \tilde{c}+\rho_{q-1});$$

$$\rho_q^* = \arg\min_{||\rho|| \le R} ||\rho - \rho_q||,$$

where $B_{p-1}$ is a batch and $\tau_q$ is a parameter of step size for PGD.

After we find $\delta_P$ with $P$ iterations and $\rho_Q$ with $Q$ iterations, we then update $\tilde{W}_c^{t-1}$ to $\tilde{W}_c^t$ by using a batched gradients. Details are given in Algorithm 1. By using this algorithm, we obtain a stable and faithful concept that can be trained in a self-supervised way on unlabeled data.

## C  EXAMPLE OF STEP 1

We apply various filters to enhance the quality and reduce the size of our concept set. The filters include:

- Removing concepts longer than 30 characters.
- Eliminating concepts that are too similar to target classes using cosine similarity in a text embedding space, with a similarity threshold of 0.85.
- Removing duplicate or synonymous concepts with cosine similarity threshold $> 0.9$.

For more information about the filters, please refer to Oikarinen et al. (2023).

In Figure 4, we show examples of the full prompt used for GPT-3 and partial output examples.

## D  MORE EXPERIMENTAL SETTINGS

### D.1  DATASETS

To evaluate the effectiveness of our approach, we conducted training using the FVLC framework on four distinct datasets: CIFAR-10, CIFAR-100, CUB, and Places365. These datasets offer a diverse range of tasks and challenges. CIFAR-10 and CIFAR-100, introduced by Krizhevsky et al. (2009), are widely used for general image classification. They provide a comprehensive set of labeled images for training and evaluation purposes. CUB, on the other hand, focuses specifically on fine-grained bird-species classification. It comprises a dataset of 5900 training samples, each

---

**Algorithm 1** Faithful Vision-Language Concept

---

1: **Input:** Weight matrix $W_F$ of the fully connected layer in a Label-Free CBM; Weight matrix $W_c$ for concept layer in a Label-Free CBM; backbone network $bf(\cdot)$; Training data $D$; parameters $R_1, R_2, k_2, k_3, \lambda_1, \lambda_2, \lambda_3$. Iterations number $T, P, Q$.
2: Initialize $\tilde{W}_c^0$ via random sampling from the standard Gaussian matrix.
3: **for** $t = 1, 2, \cdots, T$ **do**
4:     Initialize $\boldsymbol{\delta}_0^*$ and $\boldsymbol{\rho}_0^*$.
5:     **for** $p = 1, 2, \cdots, P$ **do**
6:         Randomly sample a batch $A_{p-1} \subset D$.
7:

$$\boldsymbol{\delta_p} = \boldsymbol{\delta_{p-1}^*} + \frac{\gamma_p}{|A_{p-1}|} \sum_{x \in A_{p-1}} \nabla_{\boldsymbol{\delta}} D(y(x, \tilde{\boldsymbol{c}}), y(x, \tilde{\boldsymbol{c}} + \boldsymbol{\delta_{p-1}^*}));$$

$$\boldsymbol{\delta_p^*} = \arg \min_{||\boldsymbol{\delta}|| \leq R} ||\boldsymbol{\delta} - \boldsymbol{\delta_p}||.$$

8:     **end for**
9:     **for** $q = 1, 2, \cdots, Q$ **do**
10:        Randomly sample a batch $B_{p-1} \subset D$.
11:

$$\boldsymbol{\rho_q} = \boldsymbol{\rho_{q-1}^*} + \frac{\tau_q}{|B_{q-1}|} \sum_{x \in B_{p-1}} \nabla_{\boldsymbol{\rho}} \mathcal{L}_{k_2}(\tilde{\boldsymbol{c}}, \tilde{\boldsymbol{c}} + \boldsymbol{\rho_{q-1}});$$

$$\boldsymbol{\rho_q^*} = \arg \min_{||\boldsymbol{\rho}|| \leq R} ||\boldsymbol{\rho} - \boldsymbol{\rho_q}||,$$

12:     **end for**
13:     Randomly sample a batch $C_t \subset D$.
14:     Update $\tilde{\boldsymbol{c}}$ using Stochastic Gradient Descent

$$\tilde{W}_c^t = \tilde{W}_c^{t-1} - \eta_t \sum_{x \in C_t} [\nabla_{\tilde{W}_c} D(y(x, \tilde{\boldsymbol{c}}), y(x, \boldsymbol{c}))\big|_{\tilde{W}_c = W_c^{t-1}} + \lambda_1 \nabla_{\tilde{W}_c} \mathcal{L}_{k_1}(\tilde{\boldsymbol{g}}(x), \boldsymbol{g}(x))\big|_{\tilde{W}_c = W_c^{t-1}}$$

$$+ \lambda_2 \nabla_{\tilde{W}_c} D(y(x, \tilde{\boldsymbol{c}}), y(x, \tilde{\boldsymbol{c}} + \boldsymbol{\delta_P^*}))\big|_{\tilde{W}_c = W_c^{t-1}} + \lambda_3 \nabla_{\tilde{W}_c} \mathcal{L}_{k_2}(\tilde{\boldsymbol{g}}(x), \tilde{\boldsymbol{g}}(x) + \boldsymbol{\rho_Q^*})\big|_{\tilde{W}_c = W_c^{t-1}}].$$

15: **end for**
16: **Return:** $\tilde{\boldsymbol{c}}^* = \tilde{\boldsymbol{c}}_T$.

---

accompanied by annotations that describe 312 concepts associated with the bird species. These concepts include attributes such as wing color (e.g., blue) and head pattern (e.g., spotted). In contrast, Places365 is primarily geared towards scene recognition. It encompasses a much larger scale, with 1-2 million training images available for use. The dataset also provides annotations that describe various concepts related to scenes. Despite the availability of concept names and annotations in the CUB and Places365 datasets, we intentionally choose not to utilize this information during the training of our FVLC models. Our objective was to demonstrate the capability of our approach to perform effectively without relying on such labels. Surprisingly, our method was able to discover similar concepts and achieve competitive performance when compared to methods that leverage the provided concept information. By training FVLC on these four datasets, which encompass general image classification (CIFAR-10/100), fine-grained bird-species classification (CUB), and scene recognition (Places365), we aimed to evaluate the effectiveness and versatility of our approach comprehensively. For other fields, such as sentiment analysis (Lai et al., 2023a;b; Hu et al., 2023a), autonomous driving (Xu et al., 2023a), and social good (Lai et al., 2023c; Li et al., 2023), we leave it in future work.

### D.2 BACKBONE

To ensure a fair and consistent comparison with previous work, we adopted specific backbone models for the CIFAR and CUB datasets, as done by Yuksekgonul et al. (2022). For CIFAR, we utilized the CLIP (RN50) image encoder as the backbone model. This choice allows us to establish a direct comparison with the results reported in the mentioned study. Regarding the CUB dataset, we employed the ResNet-18 architecture trained on CUB from imgclsmob, which has been widely used

List the most important features for recognizing something as a **horse**:
- **Four legs**
- **Mane and tail**
- **Hooves**
- **Large, muscular body**

List the things most commonly seen around a **horse**:
- **Saddle**
- **Bridle**
- **Hay bales**
- **Water trough**

Give superclasses for the word **horse**:
- **Mammal**
- **Animal**
- **Vertebrate**
- **Chordate**

List the most important features for recognizing something as a **dog**:
- **Fur or hair**
- **Pointed ears**
- **Snout or muzzle**
- **Tail**

List the things most commonly seen around a **dog**:
- **Leashes or collars**
- **Dog food or treats**
- **Toys, such as balls or chew toys**
- **Dog beds or blankets**

Give superclasses for the word **dog**:
- **Living organism**
- **Animal**
- **Natural object**
- **Vertebrate**

Figure 4: Example of our Step 1.

in the literature for fine-grained bird-species classification tasks, making it a suitable choice for our evaluation. For Places365, we opted for the ResNet-50 architecture trained as our backbone network. This choice aligns with common practices in the field, ensuring consistency and allowing for meaningful comparisons with other approaches. In our framework, the number of concepts incorporated into each model is roughly proportional to the number of output classes specific to the task. As each class introduces additional initial concepts, the number of concepts increases accordingly. For instance, the CIFAR-10 model integrates 128 concepts, while the CIFAR-100 model employs 824 concepts. The CUB model incorporates 211 concepts, whereas the Places365 model incorporates 2202 concepts. It is worth noting that the CUB dataset has a smaller number of concepts compared to the other datasets. This is because, in our approach, we only utilized the important features prompt for this particular dataset. Despite this reduction in the number of concepts, our method still achieves competitive performance on CUB, showcasing the effectiveness of our approach even with a more limited concept space.

## D.3 BASELINE

**Standard.** The standard model represents an image classification model, which extracts image features through the same backbone as our FVLC model and then connects a fully connected layer to complete the image classification task. In fact, this is a common practice in the field of CBM. After adding the bottleneck layer, the accuracy of the image classification task will be affected due to the intervention of the concept task.

**P-CBM.** The Post-hoc Concept Bottleneck Model (P-CBM) is a framework that converts a pre-trained neural network into a concept bottleneck model. The PCBM consists of two main steps. First, there is a pre-trained backbone model that maps inputs to an embedding space. This backbone model can be any pre-trained model, such as an image encoder or a specific layer of a deep neural network. The second step involves learning a concept bank, which captures interpretable concepts relevant to the task. Concept representations can be obtained using techniques like Concept Activation Vectors (CAVs) or multimodal models. CAVs are learned by training Support Vector Machines (SVMs) to distinguish embeddings with and without the concept. Multimodal models leverage text encoders to map concepts to vector representations. Once the concept bank is obtained, the embeddings produced by the backbone model are projected onto the concept subspace defined by the concept vectors. This projection step aligns the embeddings with the interpretable concepts, creating a bottleneck representation. Finally, an interpretable predictor is trained to classify examples based on their projections onto the concept subspace. This predictor can be implemented using methods like sparse linear layers.

### D.4 SETTINGS

To give the details of our experimental setup, we provided Table 5, which lists the key parameters we used during our training and evaluation process. The values of these parameters have been selected based on previous research and experimental experience, and have been carefully adjusted to achieve optimal performance. Note that these parameters include not only model architecture and optimizer type, but also important settings such as learning rate, batch size, number of training iterations, etc. Through Table 5, the reader can understand the specific configuration of our experiment and can reproduce it if necessary.

| Argument | Value | Remark |
|---|---|---|
| batch_size | 512 | batch size used when saving model/CLIP activations |
| saga_batch_size | 256 | batch size used when fitting final layer |
| proj_batch_size | 5000 | batch size to use when learning projection layer |
| clip_cutoff | 0.25 | concepts with smaller top5 clip activation will be deleted |
| proj_steps | 1000 | how many steps to train the projection layer for |
| interpretability_cutoff | 0.45 | concepts with smaller similarity to target concept will be deleted |
| lam | 0.0007 | sparsity regularization parameter, higher-¿more sparse |
| n_iters | 1000 | how many iterations to run the final layer solver for |
| pgd_radius | 0.1 | |
| pgd_step | 10 | |
| pgd_step_size | 0.02 | |
| pgd_norm_type | l-infty | |
| x_pgd_radius | 0.05 | |
| x_pgd_step | 10 | |
| x_pgd_step_size | 0.01 | |
| x_pgd_norm_type | l-infty | |
| lambda_1 | 1.00E-02 | |
| lambda_2 | 1.00E-02 | |
| lambda_3 | 1.00E-02 | |
| lambda_4 | 1.00E-02 | |

Table 5: Model parameter configuration.

## E DETAIL OF FIGURE 3

See in Figure 5,6,7,8.

## F MORE EXPERIMENTS

In order to verify that FVLC is faithful under the same concept space, we repeatedly generated concept sets to build different concept spaces and repeated the above experiments. The experimental results are shown in the Table 6, 7, 8, 9, 10, 11. According to the experimental results, it can be seen that our FVLC is more stable than other baselines against input perturbation and concept set perturbation, making it a more faithful interpretation. Moreover, our approach exhibits minimal accuracy degradation compared to vanilla CBM.

## G MORE EXPERIMENTS IN HAM10000

We report the performance of our model on the HAM10000 dataset in Table 12. We also visualized interpretable output results for samples in HAM10000. The presentations are shown in the Figure 9.

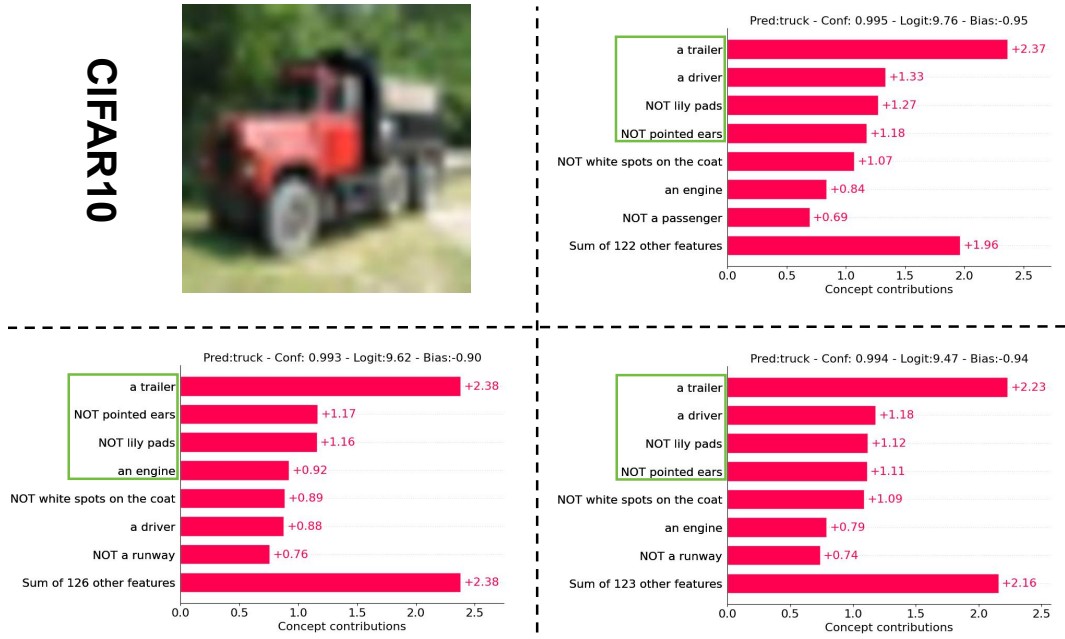

Figure 5: The visualizations for concept weights and final layer weights on one sample from CI-FAR10. Top left, top right, bottom left and bottom right are the input image, the concept weight visualization without perturbation, the concept weight visualization with perturbation, and the optimized concept weight visualization with perturbation.

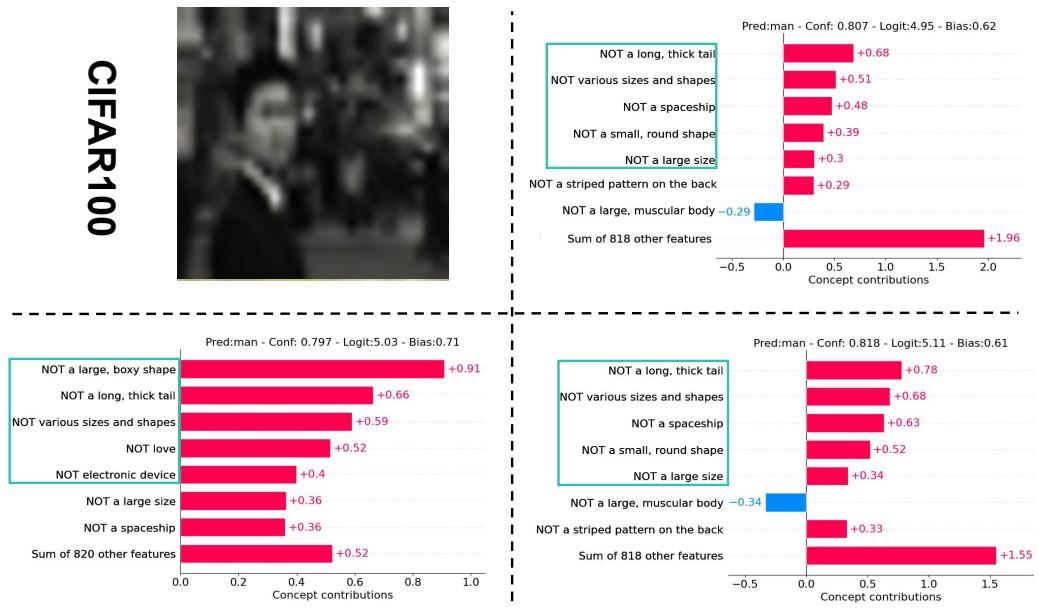

Figure 6: The visualizations for concept weights and final layer weights on one sample from CI-FAR100. Top left, top right, bottom left and bottom right are the input image, the concept weight visualization without perturbation, the concept weight visualization with perturbation, and the optimized concept weight visualization with perturbation.

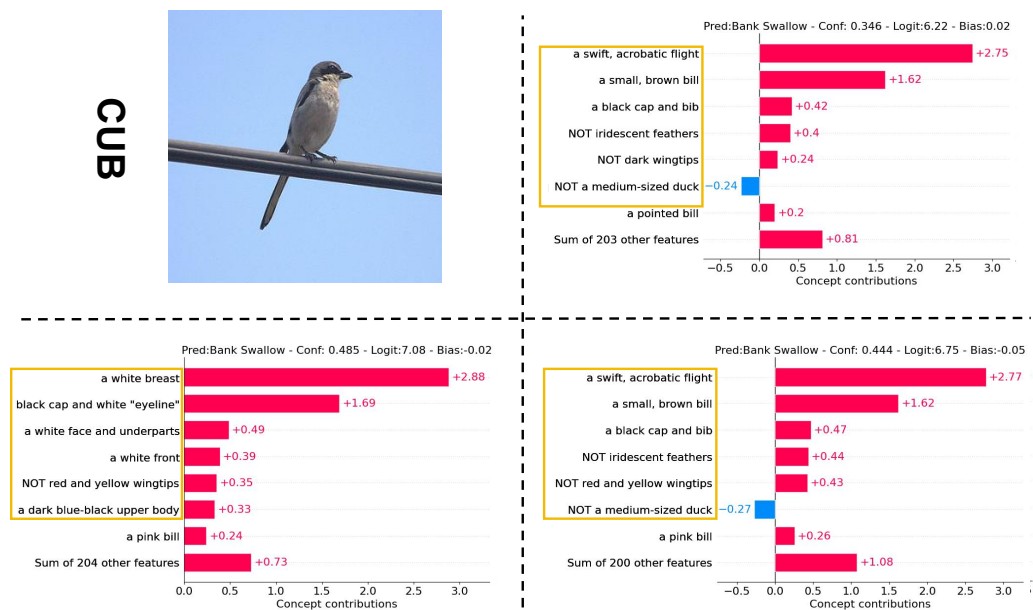

Figure 7: The visualizations for concept weights and final layer weights on one sample from CUB. Top left, top right, bottom left and bottom right are the input image, the concept weight visualization without perturbation, the concept weight visualization with perturbation, and the optimized concept weight visualization with perturbation.

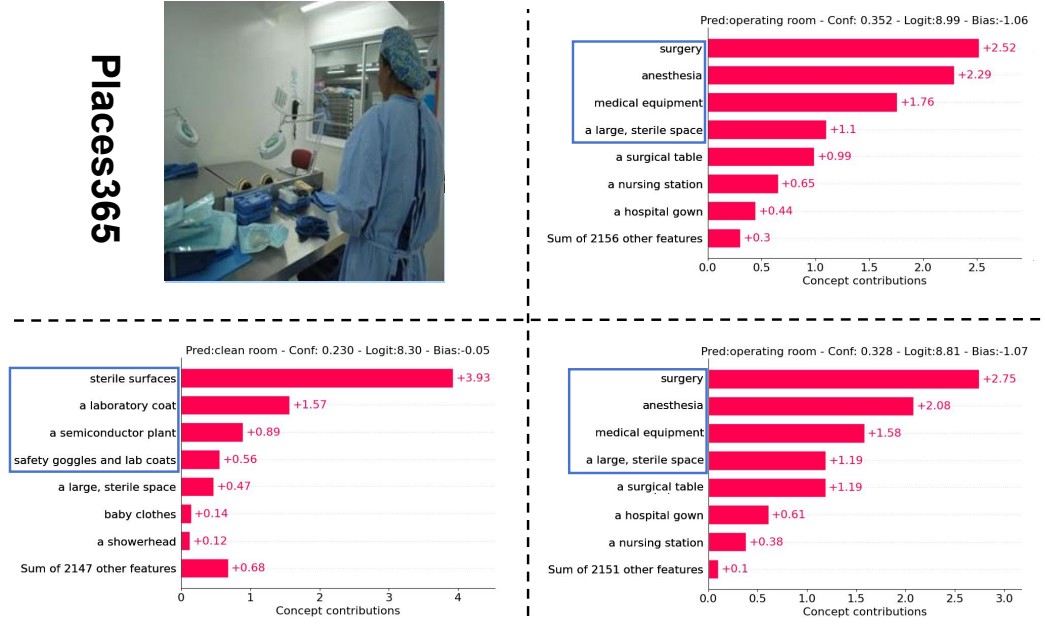

Figure 8: The visualizations for concept weights and final layer weights on one sample from Places365. Top left, top right, bottom left and bottom right are the input image, the concept weight visualization without perturbation, the concept weight visualization with perturbation, and the optimized concept weight visualization with perturbation.

| Method | CIFAR10 | CIFAR100 | CUB | Places365 |
|---|---|---|---|---|
| Standard(No interpretability) | 88.80% | 70.10% | 76.70% | 48.56% |
| P-CBM(CLIP) | 84.50% | 56.00% | N/A | N/A |
| Label-free CBM | 86.32% | 65.42% | 74.23% | 43.63% |
| WP1(5%) - base | 86.26% | 65.25% | 73.92% | 43.52% |
| WP1(5%) - **FVLC** | 86.45% | 65.59% | 73.74% | 43.47% |
| WP1(10%) - base | 85.88% | 65.05% | 74.27% | 43.49% |
| WP1(10%) - **FVLC** | 86.27% | 64.65% | 73.66% | 43.46% |
| WP2 - base | 86.09% | 65.27% | 73.62% | 43.42% |
| WP2 - **FVLC** | 86.41% | 65.52% | **74.55%** | 44.56% |
| IP - base | 86.31% | **65.67%** | 74.51% | 43.58% |
| IP - **FVLC** | **86.70%** | 65.46% | 74.11% | **43.76%** |
| WP1(5%)+WP2 - base | 86.22% | 65.18% | 73.78% | 43.66% |
| WP1(5%)+WP2 - **FVLC** | 86.50% | 65.65% | 73.66% | 43.63% |
| WP1(10%)+WP2 - base | 86.21% | 65.11% | 73.63% | 43.63% |
| WP1(10%)+WP2 - **FVLC** | 86.60% | 65.18% | 74.19% | 43.60% |
| WP1(10%)+WP2+IP - base | 85.63% | 64.72% | 74.05% | 43.20% |
| WP1(10%)+WP2+IP - **FVLC** | 86.48% | 65.28% | 74.42% | 43.33% |

Table 6: The table shows the accuracy of baseline and FVLC before and after perturbation under four benchmark datasets. The first is the standard backbone image classification model without interpretability. P-CBM (Yuksekgonul et al., 2022) and Label-free CBM (Oikarinen et al., 2023) are the latest CBM models with interpretability, and the process of generating concept sets is unsupervised and does not require manual labeling. The percentages in parentheses are the degree of added WP1. It should be emphasized that this is a repeated experiment under the new-1 concept space.

| Method | CIFAR10 | CIFAR100 | CUB | Places365 |
|---|---|---|---|---|
| Standard(No interpretability) | 88.80% | 70.10% | 76.70% | 48.56% |
| P-CBM(CLIP) | 84.50% | 56.00% | N/A | N/A |
| Label-free CBM | 86.32% | 65.42% | 74.23% | 43.63% |
| WP1(5%) - base | 86.65% | **65.45%** | 74.44% | 43.40% |
| WP1(5%) - **FVLC** | 86.01% | 65.56% | 73.83% | 43.60% |
| WP1(10%) - base | 86.20% | 65.08% | 73.62% | 43.51% |
| WP1(10%) - **FVLC** | 86.19% | 64.72% | 73.75% | 43.77% |
| WP2 - base | 86.08% | 64.89% | 74.09% | 43.63% |
| WP2 - **FVLC** | 86.31% | 65.07% | 74.65% | **44.73%** |
| IP - base | 86.88% | 65.43% | 74.49% | 43.59% |
| IP - **FVLC** | **86.95%** | 65.37% | 73.98% | 43.90% |
| WP1(5%)+WP2 - base | 86.45% | 65.20% | 73.71% | 43.80% |
| WP1(5%)+WP2 - **FVLC** | 86.76% | 65.12% | 74.21% | 43.56% |
| WP1(10%)+WP2 - base | 86.11% | 64.89% | 74.18% | 43.51% |
| WP1(10%)+WP2 - **FVLC** | 86.28% | 65.31% | 73.89% | 43.46% |
| WP1(10%)+WP2+IP - base | 86.08% | 64.59% | 74.03% | 43.12% |
| WP1(10%)+WP2+IP - **FVLC** | 86.86% | 64.87% | **74.71%** | 43.38% |

Table 7: The table shows the accuracy of baseline and FVLC before and after perturbation under four benchmark datasets. The first is the standard backbone image classification model without interpretability. P-CBM (Yuksekgonul et al., 2022) and Label-free CBM (Oikarinen et al., 2023) are the latest CBM models with interpretability, and the process of generating concept sets is unsupervised and does not require manual labeling. The percentages in parentheses are the degree of added WP1. It should be emphasized that this is a repeated experiment under the new-2 concept space.

| Method | CIFAR10 | CIFAR100 | CUB | Places365 |
|---|---|---|---|---|
| Standard(No interpretability) | 88.80% | 70.10% | 76.70% | 48.56% |
| P-CBM(CLIP) | 84.50% | 56.00% | N/A | N/A |
| Label-free CBM | 86.32% | 65.42% | 74.23% | 43.63% |
| WP1(5%) - base | 86.29% | 65.06% | 73.81% | 43.74% |
| WP1(5%) - **FVLC** | 85.99% | **65.59%** | 74.12% | 43.53% |
| WP1(10%) - base | 85.91% | 65.14% | 73.82% | 43.83% |
| WP1(10%) - **FVLC** | 86.02% | 64.83% | 73.74% | 43.70% |
| WP2 - base | 86.40% | 65.36% | 74.04% | 43.44% |
| WP2 - **FVLC** | 86.32% | 65.51% | **74.70%** | **44.48%** |
| IP - base | 86.46% | 65.36% | 74.66% | 43.64% |
| IP - **FVLC** | 86.75% | 65.60% | 74.11% | 43.74% |
| WP1(5%)+WP2 - base | **86.90%** | 65.40% | 74.18% | 43.52% |
| WP1(5%)+WP2 - **FVLC** | 86.57% | 65.20% | 73.89% | 43.53% |
| WP1(10%)+WP2 - base | 86.25% | 64.72% | 73.82% | 43.65% |
| WP1(10%)+WP2 - **FVLC** | 86.53% | 65.20% | 73.75% | 43.34% |
| WP1(10%)+WP2+IP - base | 86.15% | 64.18% | 73.85% | 43.43% |
| WP1(10%)+WP2+IP - **FVLC** | 86.89% | 65.29% | 74.12% | 43.56% |

Table 8: The table shows the accuracy of baseline and FVLC before and after perturbation under four benchmark datasets. The first is the standard backbone image classification model without interpretability. P-CBM (Yuksekgonul et al., 2022) and Label-free CBM (Oikarinen et al., 2023) are the latest CBM models with interpretability, and the process of generating concept sets is unsupervised and does not require manual labeling. The percentages in parentheses are the degree of added WP1. It should be emphasized that this is a repeated experiment under the new-3 concept space.

| Method | CIFAR10 | | CIFAR100 | | CUB | | Places365 | |
|---|---|---|---|---|---|---|---|---|
| | TCPC | TOPC | TCPC | TOPC | TCPC | TOPC | TCPC | TOPC |
| WP1(5%) - base | 1.51E-01 | 6.38E-02 | 1.04E-01 | 6.92E-02 | 1.28E-01 | 1.76E-01 | 1.65E-01 | 6.61E-02 |
| WP1(5%) - **FVLC** | 1.11E-03 | 8.96E-03 | 2.77E-03 | 4.38E-03 | 1.08E-02 | 1.46E-03 | 1.44E-03 | 1.25E-03 |
| WP1(10%) - base | 2.06E-01 | 8.51E-02 | 1.92E-01 | 1.32E-01 | 2.39E-01 | 3.37E-01 | 2.16E-01 | 1.15E-01 |
| WP1(10%) - **FVLC** | 1.17E-03 | 7.07E-03 | 3.85E-03 | 4.34E-03 | 1.22E-02 | 1.55E-03 | 1.33E-03 | 1.25E-03 |
| WP2 - base | 1.60E-01 | 4.77E-02 | 1.33E-01 | 6.53E-02 | 1.38E-01 | 1.71E-01 | 1.43E-01 | 6.10E-02 |
| WP2 - **FVLC** | 1.06E-02 | 8.67E-03 | 3.41E-03 | 4.55E-03 | 1.07E-02 | 1.60E-03 | 1.56E-03 | 1.35E-03 |
| IP - base | 1.70E-01 | 6.02E-02 | 1.35E-01 | 8.87E-02 | 1.76E-01 | 2.19E-01 | 1.65E-01 | 7.94E-02 |
| IP - **FVLC** | 7.82E-03 | 8.43E-03 | 3.30E-03 | 4.46E-03 | 1.03E-02 | 1.55E-03 | 1.53E-03 | 1.27E-03 |
| WP1(5%)+WP2 - base | 1.78E-01 | 3.60E-02 | 1.24E-01 | 6.58E-02 | 1.46E-01 | 1.86E-01 | 1.64E-01 | 6.36E-02 |
| WP1(5%)+WP2 - **FVLC** | 1.25E-02 | 7.76E-03 | 3.84E-03 | 4.41E-03 | 9.80E-02 | 1.56E-03 | 1.58E-03 | 1.25E-03 |
| WP1(10%)+WP2 - base | 1.21E-01 | 8.50E-02 | 1.93E-01 | 1.28E-01 | 1.76E-01 | 3.32E-01 | 2.53E-01 | 1.12E-01 |
| WP1(10%)+WP2 - **FVLC** | 1.14E-02 | 9.78E-03 | 2.11E-02 | 1.45E-02 | 1.94E-02 | 3.88E-02 | 2.68E-02 | 1.21E-02 |
| WP1(10%)+WP2+IP - base | 1.39E-01 | 1.03E-01 | 2.32E-01 | 1.56E-01 | 1.94E-01 | 3.57E-01 | 2.55E-01 | 1.42E-01 |
| WP1(10%)+WP2+IP - **FVLC** | 1.41E-02 | 1.15E-02 | 2.47E-02 | 1.75E-02 | 2.13E-02 | 4.70E-02 | 3.44E-02 | 1.28E-02 |

Table 9: The table shows the TCPC and TOPC of baselines and FVLC under four benchmark datasets with different perturbations. The percentages in parentheses are the degree of added WP1. It should be emphasized that this is a repeated experiment under the new-1 concept space.

| Method | CIFAR10 | | CIFAR100 | | CUB | | Places365 | |
|---|---|---|---|---|---|---|---|---|
| | TCPC | TOPC | TCPC | TOPC | TCPC | TOPC | TCPC | TOPC |
| WP1(5%) - base | 1.47E-01 | 6.19E-02 | 1.07E-01 | 7.10E-02 | 1.22E-01 | 1.82E-01 | 1.63E-01 | 6.77E-02 |
| WP1(5%) - **FVLC** | 1.07E-03 | 8.95E-03 | 2.79E-03 | 4.31E-03 | 1.09E-02 | 1.40E-03 | 1.46E-03 | 1.26E-03 |
| WP1(10%) - base | 2.12E-01 | 8.40E-02 | 1.84E-01 | 1.27E-01 | 2.37E-01 | 3.44E-01 | 2.09E-01 | 1.10E-01 |
| WP1(10%) - **FVLC** | 1.13E-03 | 6.73E-03 | 3.84E-03 | 4.36E-03 | 1.16E-02 | 1.56E-03 | 1.39E-03 | 1.24E-03 |
| WP2 - base | 1.61E-01 | 4.74E-02 | 1.36E-01 | 6.23E-02 | 1.33E-01 | 1.68E-01 | 1.46E-01 | 6.04E-02 |
| WP2 - **FVLC** | 1.05E-02 | 8.57E-03 | 3.36E-03 | 4.60E-03 | 1.08E-02 | 1.67E-03 | 1.54E-03 | 1.35E-03 |
| IP - base | 1.68E-01 | 5.93E-02 | 1.37E-01 | 8.54E-02 | 1.69E-01 | 2.19E-01 | 1.71E-01 | 7.98E-02 |
| IP - **FVLC** | 8.04E-03 | 8.04E-03 | 3.29E-03 | 4.34E-03 | 1.05E-02 | 1.56E-03 | 1.49E-03 | 1.26E-03 |
| WP1(5%)+WP2 - base | 1.87E-01 | 3.43E-02 | 1.20E-01 | 6.33E-02 | 1.53E-01 | 1.87E-01 | 1.61E-01 | 6.33E-02 |
| WP1(5%)+WP2 - **FVLC** | 1.22E-02 | 8.12E-03 | 3.85E-03 | 4.34E-03 | 9.40E-02 | 1.54E-03 | 1.59E-03 | 1.22E-03 |
| WP1(10%)+WP2 - base | 1.20E-01 | 8.33E-02 | 2.01E-01 | 1.33E-01 | 1.68E-01 | 3.43E-01 | 2.53E-01 | 1.12E-01 |
| WP1(10%)+WP2 - **FVLC** | 1.12E-02 | 9.32E-03 | 2.14E-02 | 1.41E-02 | 2.01E-02 | 3.72E-02 | 2.65E-02 | 1.17E-02 |
| WP1(10%)+WP2+IP - base | 1.40E-01 | 9.84E-02 | 2.38E-01 | 1.50E-01 | 1.87E-01 | 3.49E-01 | 2.63E-01 | 1.35E-01 |
| WP1(10%)+WP2+IP - **FVLC** | 1.35E-02 | 1.17E-02 | 2.59E-02 | 1.83E-02 | 2.10E-02 | 4.86E-02 | 3.45E-02 | 1.28E-02 |

Table 10: The table shows the TCPC and TOPC of baseline and FVLC under four benchmark datasets with different perturbations. The percentages in parentheses are the degree of added WP1. It should be emphasized that this is a repeated experiment under the new-2 concept space.

| Method | CIFAR10 | | CIFAR100 | | CUB | | Places365 | |
|---|---|---|---|---|---|---|---|---|
| | TCPC | TOPC | TCPC | TOPC | TCPC | TOPC | TCPC | TOPC |
| WP1(5%) - base | 1.44E-01 | 6.44E-02 | 1.00E-01 | 6.61E-02 | 1.25E-01 | 1.81E-01 | 1.72E-01 | 6.58E-02 |
| WP1(5%) - **FVLC** | 1.12E-03 | 8.58E-03 | 2.68E-03 | 4.50E-03 | 1.06E-02 | 1.51E-03 | 1.42E-03 | 1.30E-03 |
| WP1(10%) - base | 2.08E-01 | 8.31E-02 | 1.85E-01 | 1.34E-01 | 2.41E-01 | 3.40E-01 | 2.26E-01 | 1.21E-01 |
| WP1(10%) - **FVLC** | 1.22E-03 | 6.73E-03 | 3.88E-03 | 4.31E-03 | 1.25E-02 | 1.53E-03 | 1.30E-03 | 1.25E-03 |
| WP2 - base | 1.58E-01 | 4.89E-02 | 1.37E-01 | 6.83E-02 | 1.42E-01 | 1.67E-01 | 1.44E-01 | 6.17E-02 |
| WP2 - **FVLC** | 1.03E-02 | 8.73E-03 | 3.24E-03 | 4.34E-03 | 1.04E-02 | 1.65E-03 | 1.55E-03 | 1.35E-03 |
| IP - base | 1.78E-01 | 6.11E-02 | 1.30E-01 | 9.07E-02 | 1.80E-01 | 2.23E-01 | 1.62E-01 | 7.94E-02 |
| IP - **FVLC** | 7.49E-03 | 8.14E-03 | 3.42E-03 | 4.28E-03 | 1.04E-02 | 1.48E-03 | 1.54E-03 | 1.22E-03 |
| WP1(5%)+WP2 - base | 1.82E-01 | 3.73E-02 | 1.30E-01 | 6.61E-02 | 1.50E-01 | 1.82E-01 | 1.63E-01 | 6.17E-02 |
| WP1(5%)+WP2 - **FVLC** | 1.25E-02 | 8.15E-03 | 3.91E-03 | 4.57E-03 | 9.85E-02 | 1.52E-03 | 1.64E-03 | 1.19E-03 |
| WP1(10%)+WP2 - base | 1.15E-01 | 8.60E-02 | 1.87E-01 | 1.26E-01 | 1.76E-01 | 3.20E-01 | 2.64E-01 | 1.07E-01 |
| WP1(10%)+WP2 - **FVLC** | 1.18E-02 | 9.70E-03 | 2.17E-02 | 1.47E-02 | 2.02E-02 | 3.74E-02 | 2.71E-02 | 1.25E-02 |
| WP1(10%)+WP2+IP - base | 1.38E-01 | 1.07E-01 | 2.22E-01 | 1.62E-01 | 2.01E-01 | 3.61E-01 | 2.62E-01 | 1.41E-01 |
| WP1(10%)+WP2+IP - **FVLC** | 1.36E-02 | 1.20E-02 | 2.39E-02 | 1.70E-02 | 2.22E-02 | 4.57E-02 | 3.50E-02 | 1.27E-02 |

Table 11: The table shows the TCPC and TOPC of baselines and FVLC under four benchmark datasets with different perturbations. The percentages in parentheses are the degree of added WP1. It should be emphasized that this is a repeated experiment under the new-3 concept space.

| Model | HAM10000 |
|---|---|
| Standard (No interpretability) | 96.30% |
| P-CBM | **94.70%** |
| Label-free CBM | 94.38% |
| FVLC | 94.42% |

Table 12: The performance of our model on the HAM10000 dataset.

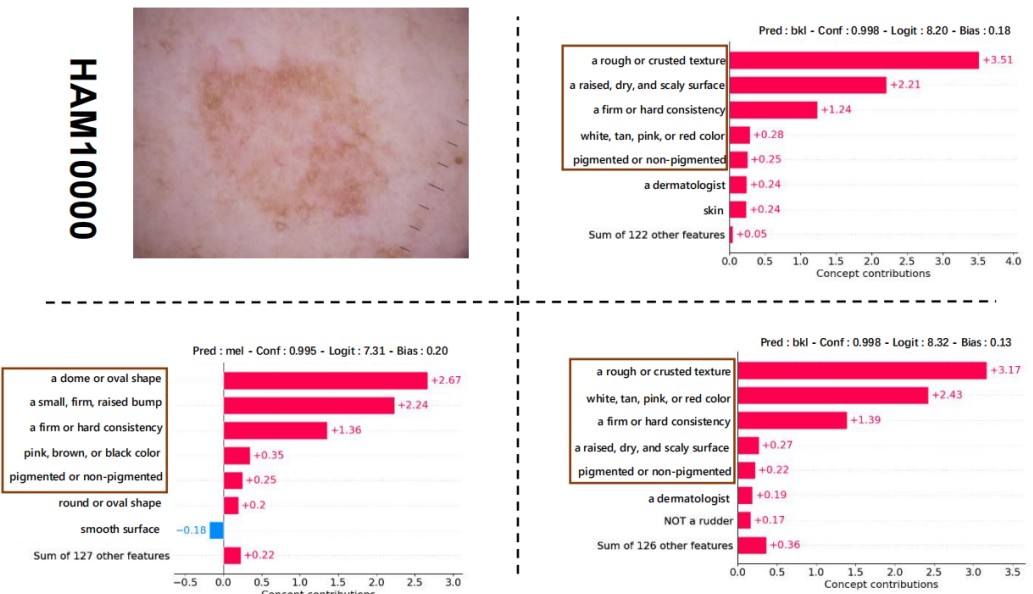

Figure 9: Explanations for one randomly chosen input image for our model trained on HAM10000. The visualizations for concept weights and final layer weights on one sample from HAM10000. Top left, top right, bottom left and bottom right are the input image, the concept weight visualization without perturbation, the concept weight visualization with perturbation, and the optimized concept weight visualization with perturbation.

