# OpenReview forum: "Faithful Vision-Language Interpretation via Concept Bottleneck Models"
_ICLR.cc/2024/Conference — ICLR 2024 poster_

### Official Review · Reviewer_W5Db · 2023-10-26

**Soundness:** 2 fair
**Presentation:** 3 good
**Contribution:** 3 good
**Rating:** 6
**Confidence:** 3

**Summary:**

This paper introduces Faithful Vision-Language Concept (FVLC) models to overcome faithfulness and instability issues in Label-free Concept Bottleneck models (LCBMs). The authors also analyze the stability and the faithfulness of concepts introducing four desiderata, namely: (i) the degree of overlap with the ground-truth concepts, (ii) the robustness against random noise, (iii) the correctness of the predictions, and (iv) the stability of the output layer.
With a series of experiments, it is shown that the FVLC outperforms LCBMs both in accuracy and in stability of the concepts.

**Strengths:**

The authors address the important problem on interpretability in CBMs. Ensuring concepts are of good quality is central to applications of CBMs and LCBMs in real-world scenarios where perturbations are likely to happen. The presentation is clear and all parts can be easily understood, although it could be improved, see weaknesses.
The analysis of the possible instability effects in the concepts is sound and motivated by examples where LCBMs effectively struggle to predict the correct concepts. The proposed method for concept refinement is simple and works well in a wide range of case studies. In addition, the ablation study reveals that all terms included in FVLC are needed to improve the stability and accuracy of the model.
Essentially, FVLC paves the use of language-based CBMs with faithfulness guarantees, which could be very helpful in applications.

**Weaknesses:**

One major concern about the analysis is the question of the interpretability. The claim that if a concept $\tilde c$ matches the top-k component of c leads to an equivalently interpretable concept depends on the assumption that c is already interpretable. This is implicit in the discussion and it seems to me that it is sidelined. Since the interpretability of the model is central in this paper, somehow this has to be verified, e.g., via concept accuracy, even for LCBMs.

The authors consider the LCBM as the standard reference for improving the concepts, but the optimization could have been done even without it since the alignment loss has to be w.r.t. concepts extracted with CLIP, avoiding the noise in fitting first the LCBM and then the FVLC.

I feel the contribution would also improve by comparing with other SotA models, like Post-hoc CBMs and LaBo:\
[1] Yuksekgonul, Mert, Maggie Wang, and James Zou, "Post-hoc concept bottleneck models." ICLR 2023 \
[2] Yang, Yue, et al. "Language in a bottle: Language model guided concept bottlenecks for interpretable image classification." CVPR 2023

About the presentation, it is somehow counterintuitive to mention $\Tilde c$ and $c$ while there is a clear reference to the maps $\Tilde g$ and $g$. Moreover, I am sure that all the extracted concepts are input-dependent, so it would be the case to indicate them with $g(x)$ or $c(x)$ to explicit their dependence.
One possible improvement of the presentation could be to comment on the definitions after their statements, indicating intuitive properties captured by them. Another point is the introduction of $ \tilde c$ which is clear from the text but it should be stated before that the authors refer to the faithful concepts obtained with $\TIlde g$.

One remark: the statement in section 5 that $\tilde c = c$ in the limit of $\alpha_1 \to 0$ is not always true since it depends on the loss that is considered. What loss is used? In principle, different choices of the loss function would to more stable or less stable concepts for conditions (iii) and (iv).

**Questions:**

How is the interpretability of the model evaluated in vision-language models?

**Details Of Ethics Concerns:**

No ethical concerns are present.

---

> ### Author Response · Authors · 2023-11-20
> **Response to W5Db**
>
> Dear Reviewer **W5Db**,
>
> We would like to express our sincere gratitude for taking the time to review our paper and providing valuable feedback. We have carefully considered your comments and have made revisions to address each of the issues raised. Please find our responses to your specific points below:
>
> **Response to W1:**
>
> Unlike the original concept model, there is no ground truth concept annotated by experts. So it is hard to evaluate the explainability. The same issue also appears in Label-free CBM so the metric of  explainability is out of the scope of this paper. In our paper, we verify the effectiveness of the output explanatory results by outputting a large number of visual samples.
>
> **Response to W2:**
>
> We kindly disagree with your point. Actually, our objective function (2) heavily depends on the Label-Free CBM:  (1) We can easily see that $y(x, \tilde{c})$ and $y(x, c)$ depend on the weight matrix $W_F$ of the fully connected layer of a trained  Label-Free CBM.  (2) Moreover, $\mathcal{L}_{k} (\tilde{g}(x), g(x))$, which is a surrogate loss of $V_k$, depends on $W_c$, which is the weight matrix of the concept layer in Label-Free CBM. We can also see these from Definition 2.  Thus, FVLC heavily relies on the performance of its corresponding Label-Free CBM.
>
> **Response to W3:**
>
> **In terms of accuracy on image classification tasks, we have compared with PCBM.** However, in the task of evaluating interpretability, although these are concept-based models, their concept outputs are different, so it is difficult to make a comparison. In addition, for recently developed unsupervised CBM models (LCBM and PCBM), it is difficult to quantify the effectiveness of their interpretability and the "accuracy" of the output concept. The goal of our work is to make unsupervised CBM models more faithful.
>
> **Response to W4:**
>
> Sorry for the confusion, yes, you are right all the extracted concepts are input-dependent. To address your concern, in the revised version, we rewrote Definition 2 and our optimization procedure in the revised version. Moreover, we give more details on the optimization procedure in Appendix B.  About intuitive properties, in the revised version, we add one additional section Appendix A on more intuition of our Definition 2 and its four properties.
>
> About notation $\tilde{c}$, we have modified it in the revised version.
>
> **Response to Remark:**
>
> Sorry for the confusion, in fact, $D$ should be a divergence or probability distance. In this case, our claim will be correct. We have changed it in the revised version. Yes, different losses may lead to different performances.  In fact, in our experiments, we consider $D$ as the KL divergence for ease of optimization.

---

> ### Comment · Reviewer_W5Db · 2023-11-21
> **Reply to authors**
>
> I thank the authors for going through my review and for the effort in improving the paper.
>
> Regarding my review, I want to comment on some points:
>
> 1) *Interpretability of Language-based CBMs* As I understand from your reply, measures of concept interpretability are not adopted in Language-based CBMs. This can be a serious limitation of these CBMs, I think guaranteeing faithfulness cannot be enough without interpretability. This should be discussed in a limitations section
>
> 2) *Using the LFCBM as a reference* Thank you for the clarification and I do not want to depreciate the method you proposed. I understand your point but still, if the FVLC improves the existing LFCBM but takes it as a reference, would it inherit limitations of LFCBM (e.g. if some predictions are wrong for the LFCBM they will be wrong also in the FVLC)?
>
> 3) *Comparison with PCBMs* Thank you for pointing to the comparison, I will adjust my confidence score.
>
> 4) I would not call PCBMs, LABO, and LFCBs unsupervised methods (like Self-explainable Neural Networks which are unsupervised), rather weakly supervised or multimodal.
>
> **Remark** If you use the KL divergence you can miss the scale due to the softmax operation for obtaining $p(y \mid c)$. MSE or l1 would be better, maybe.

---

> > ### Author Response · Authors · 2023-11-21
> >
> > Dear Reviewer,
> >
> > Thanks for your comments.
> >
> >
> > (1) Yes, we think this is a limitation for language-based CBMs and it is indeed out of the scope of this. This paper aims to improve the stability of Label-free CBM
> >
> >
> > (2) Yes, we agree that FVLC inherits the advantages and disadvantages of Label-free CBM. However, here we focus on Label-free CBM because Label-free CBM current is the SOTA for CBM without concepts. Although FVLC is for Label-free CBM, its general intuition is natural (see Appendix A) and it can be extended to other future advanced CBMs.
> >
> >
> > (4) and remark Thanks for pointing out these
> >
> > We are happy to discuss more if there are any additional concerns. We are looking forward to your feedback and would greatly appreciate you consider raising the scores.
> >
> > Thank you,
> >
> > Authors

---

### Official Review · Reviewer_6N2M · 2023-10-29

**Soundness:** 4 excellent
**Presentation:** 3 good
**Contribution:** 3 good
**Rating:** 6
**Confidence:** 3

**Summary:**

In this paper, the author proposed a solution for constructing faithful vision-language concept models. To achieve this, the authors formalized four properties that faithful concepts should respect and designed a min-max object function by following these four properties. In addition, the authors also proposed an algorithm to solve this object function. Through extensive experiments over multiple vision-language dataset, the authors can demonstrate the effectiveness of the proposed solution in maintaining good prediction performance while achieving faithfulness.

**Strengths:**

+ Label-free concept bottleneck model can be unstable and unfaithful. The authors provide a timely solution for tackling this problem
+ The authors provide a very clear formulation of the faithful concept bottleneck model and the overall techniques are sound and convincing
+ The authors also performed extensive experiments over various vision-language datasets to demonstrate the benefit of the proposed solution
+ The overall presentation is clear and easy to follow.

**Weaknesses:**

+ I have one concern for the generalizability of the proposed solution. Although the authors claimed that they proposed a solution to deal with the stability issue of the label-free concept bottleneck model, the authors only focused on the prediction task over the vision dataset. I am wondering whether the proposed solution could be generalized to other settings such as the classification tasks over NLP data or tabular data.
+ The metrics of evaluating the model stability are also concerning to me. I feel that this metric should be $\epsilon$-invariant, which means that ideally the perturbation $\epsilon$ should occur in the denominator of the metric. Otherwise, with larger epsilon, you can get larger stability, which seems to be counter-intuitive. Also, by looking at the other similar metrics in literature, such as the stability metric proposed in the model interpretability literature (Alvarez Melis, David, and Tommi Jaakkola. "Towards robust interpretability with self-explaining neural networks." Advances in neural information processing systems 31 (2018).), they are also calculated by putting the input perturbation in the denominator. So it would be better if the authors could justify why they didn't choose to design their stability metric in this manner and ideally reference some other papers.
+ The presentation could be further improved. For example, for Figure 3, the text in the figure is very tiny and it is very hard to tell what concepts they are. For Figure 2, I guess the box mixed with black and white dots should represent the noise but I cannot find any explanations for this. It would be better if the authors could clearly illustrate this figure in the caption.

**Questions:**

See above

---

> ### Author Response · Authors · 2023-11-20
> **Response to 6N2W**
>
> Dear Reviewer **6N2M**,
>
> We sincerely appreciate the time you took to review our paper and provide valuable feedback. We have carefully considered your comments and made revisions to address each issue raised. Please find our responses to your specific points below:
>
> **Response to W1:**
> Briefly speaking, the main idea and our faithful concept can be directly extended to other forms of data such as text since they do not rely on the specific form of images. However, we are not sure about the performance of other data and this could left for future work. In fact, we think this should not be a reason that our paper will be rejected as to the best of our knowledge, most of the previous work on concept bottleneck models only considers image data and there is less study on other data types [1,2,3]. But it is certain that our method can migrate data to the remaining modes. Because we are based on the output of the feature layer for subsequent operations.
> [1] Yang, Yue, et al. "Language in a bottle: Language model guided concept bottlenecks for interpretable image classification." Proceedings of the IEEE/CVF Conference on Computer Vision and Pattern Recognition. 2023.
> [2] Chauhan, Kushal, et al. "Interactive concept bottleneck models." Proceedings of the AAAI Conference on Artificial Intelligence. Vol. 37. No. 5. 2023.
> [3] Yuksekgonul, Mert, Maggie Wang, and James Zou. "Post-hoc concept bottleneck models." arXiv preprint arXiv:2205.15480 (2022).
>
> **Response to W2:**
> Thanks for your question. There are mainly two reasons we do not use invariant metrics: (1) Note that unlike the previous paper (such as the one you mentioned) which only considers adding noises to image inputs as the perturbation, in our experiments, we consider (a combination of) three different perturbations. It is hard to normalize these three perturbations on the same scale, such as the word perturbation.  (2) Note that in our experiments, due to different combinations of perturbations, we do not intend to discuss the change of stability w.r.t. perturbation scale. Our main goal is to show that under the same scale of perturbation, our FVLC is more stable than baselines (Table 2). Thus, there is no need to consider the invariant metrics.
>
> **Response to W3:**
> Thanks for your suggestion. We will make the text in Figure 3 larger in the final version if additional pages are allowed. Moreover, in the revised version, we enlarge the fonts in Figure 3, see Figure 5-8 in Appendix for details.
> In the revised version, we also added more details to all figures (including Figure 2) in our paper.

---

> > ### Author Response · Authors · 2023-11-21
> >
> > Dear Reviewer,
> >
> > Thank you so much for your time and efforts in reviewing our paper. We have addressed your comments in detail and are happy to discuss more if there are any additional concerns. We are looking forward to your feedback and would greatly appreciate you consider raising the scores.
> >
> > Thank you,
> >
> > Authors

---

> > > ### Comment · Reviewer_6N2M · 2023-11-21
> > >
> > > Thanks very much for the authors' responses. Some of my concerns have been addressed. However, in terms of my questions on the applicability of the proposed methods to other data modalities, such as NLP data, I still feel that the authors could at least perform some preliminary results on those modalities. For example, the authors could consider the NLP applications discussed in this paper: https://arxiv.org/abs/2205.07237. But I also don't mean to reject this paper for just ignoring such experiments. It would be better if the authors could include such additional experiments in future versions.

---

### Official Review · Reviewer_UywT · 2023-10-30

**Soundness:** 3 good
**Presentation:** 2 fair
**Contribution:** 3 good
**Rating:** 5
**Confidence:** 4

**Summary:**

The concept bottleneck models(CBMs) and the subsequent label free CBMs offer the concept based explanation for a given prediction. However, their explanations are not faithful if the input image or the concept gets perturbed. The authors tried to fill this gap.

**Strengths:**

Concept based explanations are a hot topic at this time and the effect on explanations based on the input perturbations is fairly unexplored. So thats a major strength of the paper.

**Weaknesses:**

1. The writing is poor. It is hard to follow.
2. The related work is not complete. The following important papers should have been included in the related work:

[1] Concept Embedding models. Zarlenga et al. Neurips 2022
[2] Interpretable Neural-Symbolic Concept Reasoning. Barberio et al. ICML 2023
[3] Entropy-based Logic Explanations of Neural Networks. Barberio et al. AAAI 2021
[4] Addressing Leakage in Concept Bottleneck Models. Havasi et. al., Neurips 2022
[5] Dividing and Conquering a BlackBox to a Mixture of Interpretable Models: Route, Interpret, Repeat. Ghosh et al. ICML 2023
[6] Distilling BlackBox to Interpretable models for Efficient Transfer Learning. Ghosh et al. MICCAI 2023
[7] Language in a Bottle: Language Model Guided Concept Bottlenecks for Interpretable Image Classification Yang et al. CVPR 2023
[8] A Framework for Learning Ante-hoc Explainable Models via Concepts Sarkar et al. CVPR 2022

3.  What is the difference b/w faithful and original concept in definition 1?
4. Label free cbm step 1, page 3. I think it will be "they propose" instead of "we propose" as label free cbm is proposed by Oikarinen et al., 2023. All the in step 1, 2, 3, 4 in sect 3 "we" should be replaced by "they"
5. I dont understand anything from fig 2. The caption should clearly state the objective of the figure.
6. The terms in the loss function is not at all clear (Page 6). What is D? is this somedistance? Also how will the two constraint \delta and \phro will be going to affect the loss and the optimization. And its connection with the Fig.2 is completely unresolved to me.
7. Can the authors use any difficult datasets like HAM10k or MIMIC-CXR to bolster their claims. Medical images will make the claims stronger because if the explanations are not stable, it will raise trust issue among the users.
8. I am confused with table 1. If the main message is to showcase that their method is more superior compared to the baselines, they should have the best cofiguration of their method and the baselines. Also they should have another table where they should conduct an ablation study to include different configurations of their method with various perturbations and compare the performance. Also, only PCBM and LCBM baselines are not enough. I would like to see the performances for LCBM and language in a bottle especially when the inputs and the concepts will be perturbed. Also thourough comparison is needed with basic CBM and some other methods for datasets where concepts are annotated. Also missing variances in table 1.
9. Also, for table 2, nothing is highlighted or bolded. With so many numbers, it is diffcult for me to see which is higher and which is lower.

*Post Rebuttal*
I saw the comments by the authors and i would like to thank them. The writing is clear and the captions are good but very long. Still some related work is missing. Importantly, i feel the datasets are fairly simple but i am increasing the score to 5.

**Questions:**

See weaknesses

---

> ### Author Response · Authors · 2023-11-20
> **Response to UywT**
>
> **(1/2)**
>
> Dear Reviewer **UywT**,
>
> We would like to express our sincere gratitude for taking the time to review our paper and providing valuable feedback. We have carefully considered your comments and have made revisions to address each of the issues raised. Please find our responses to your specific points below:
>
> **Response to W1:**
>
> In the revised versions, to make the paper more readable, first, we added more details to each figure to provide better illustration (such as Figure 2); secondly, we highlighted the results of our method in all tables; third, we added more details on the baseline methods (Section D in Appendix); finally, we also add more intuition of our definition of FVLC, and more details on the definition of FVLC (Section A in Appendix). You can also see our response to all reviewers.
>
> We hope these changes will make you satisfied. If not, please feel free to reply to us.
>
> **Response to W2:**
>
> Thanks, we have added these references to the related work section in the revised version.
>
> **Response to W3:**
>
> **Generally speaking, our faithful concept is more stable on prediction and interpretability against different perturbations than the original concept in Label-free CBM, while it also keeps the prediction performance and interpretability of the original concept.** To give a better illustration, in Appendix A of the revised version, we provided more details on the intuition of our definition 2. Next, we will show more of the difference between faithful and original concepts in definition 1.
>
> (1) The first condition ensures that the faithful concept keeps almost the same interpretability as in the original concept for each input. This is modeled via the top-k indices overlap between these two vectors is large.
>
> (2) The second condition ensures that our faithful concept is stable for explainability w.r.t. different perturbations, especially input perturbation and word perturbation. This is significantly different from the original concept. As illustrated in Figure 1, the interpretability of the original concept is very unstable.
>
> (3) The third condition ensures that the faithful concept keeps almost the same prediction output as in the original concept for each input. This is modeled via the distance or divergence between the prediction distributions of the faithful concept and the original concept.
>
> (4) The last condition guarantees that our faithful concept is stable for the prediction w.r.t. different perturbations. This is also motivated by the fact that the original concept may have wrong predictions with perturbations.
>
> **Response to W4:**
>
> We have corrected it in section 3.
>
> **Response to W5:**
>
> In the revised version, we have added more details on all Figures including Figure 2.
>
> **Response to W6:**
>
> $D$ should be a divergence distance, which is defined in Definition 2. In our experiments, we use KL divergence.  For the role of two constraints on $\delta$ and $\rho$, please refer to Stability and Prediction on Page 5. In the revised version, we have added more details for our optimization loss. Moreover, we polished Appendix Section B with details on the optimization procedure for the relaxed problem.
>
> **Response to W7:**
>
> We have added some experiments on the HAM10000 dataset to improve the reliability of our model, as detailed in Appendix G, Table 12 and Figure 9 in the revised version. We hope these changes will make you satisfied. If not, please feel free to reply to us.
>
> **Response to W8:**
>
> (1) For more details of the experiment, such as baseline, etc., we have made additional supplements, which are detailed in Table 5 of Appendix D. We hope these changes will make you satisfied. If not, please feel free to reply to us.
>
> (2) We conducted a set of ablation experiments to compare the performance, and we did not seem to find that our optimization had a significant impact on the experimental image classification performance, but it could effectively improve the stability of the output concept (see in Table 2 and Table 3). Due to time constraints, we only conducted supplementary ablation experiments on the CIFAR10 dataset (WP1(10%)+WP2+IP - FVLC).
> | L2   | L3   | L4   | CIFAR10 |
> | ---- | ---- | ---- | ------- |
> |      |      |      | 86.69%  |
> | √    |      |      | 87.70%  |
> |      | √    |      | 87.73%  |
> |      |      | √    | 86.67%  |
> | √    | √    |      | 86.73%  |
> | √    |      | √    | 86.70%  |
> |      | √    | √    | 86.71%  |
> | √    | √    | √    | 86.70%  |
>
> (3) Although they are both concept based models, their frameworks are very different, as are the output concepts and the forms of concept weights. It is meaningless to discuss the anti-interference ability of the concept that interpretability outputs differ between models. So we only compared the performance of the model, and we also compared the performance of P-CBM, LCBM and our model.

---

> ### Author Response · Authors · 2023-11-20
> **Response to UywT**
>
> **(2/2)**
>
> (3) It is notable that it is impossible to compare our method with previous methods with annotated concepts: (1) Our concept set is generated by LLM with previous work needs to use concept sets given by experts. Since the concept sets are different, they are incomparable. (2) It is notable that the main motivation of our methods is label-free CBM so there is no ground truth of concepts. And it is meaningless to compare with previous methods.
>
> (4) We repeated the last group of experiments in Table 1 for 5 times, and calculated the mean and variance of the output data in this group as shown below. The experimental results show that the variances of our experimental data is very small. Because of time, we did not repeat all the experiments to calculate the variances. We hope these changes will be satisfactory to you. If not, please feel free to reply us with more experiments and explanations.
> | CIFAR10 | CIFAR100 | CUB | Places365 |
> | ------- | ------- | ------- | ------- |
> | 86.82%(2.15E-06) | 65.15%(1.11E-06) | 74.30%(5.03E-07) | 43.67%(2.19E-05) |
>
> **Response to W9:**
>
> In the revised version, we have highlighted all the results of our method in grey color in all tables. Please check the revision.

---

> > ### Author Response · Authors · 2023-11-21
> >
> > Dear Reviewer,
> >
> > Thank you so much for your time and efforts in reviewing our paper. We have addressed your comments in detail and are happy to discuss more if there are any additional concerns. We are looking forward to your feedback and would greatly appreciate you consider raising the scores.
> >
> > Thank you,
> >
> > Authors

---

> ### Author Response · Authors · 2023-11-22
>
> Dear Reviewer **UywT**,
>
> We would like to express our gratitude for your thorough review and the insightful comments you provided.
>
> We have carefully addressed each of your comments and suggestions in response and  revised version we submitted earlier. We believe that the revisions we made have significantly strengthened the paper and addressed the concerns you raised. If there are any remaining questions or concerns, we would be more than happy to response.
>
> Additionally, we would like to take this opportunity to request your consideration for a potential increase in the score of our paper. We believe that the revisions we made, as outlined in the rebuttal, have substantially improved the quality and contribution of our work. We hope that you will take these revisions into account and consider adjusting the score accordingly. **We are looking forward to your timely feedback and would greatly appreciate you consider raising the scores.**
>
> Thanks,
>
> Authors

---

### Official Review · Reviewer_AK8W · 2023-11-03

**Soundness:** 4 excellent
**Presentation:** 3 good
**Contribution:** 4 excellent
**Rating:** 8
**Confidence:** 4

**Summary:**

In this paper, the authors have introduced a new approach to ensure the stability and robustness of label-free concept bottleneck models. The authors introduced stability conditions for the concepts, similar to those in (Sundararajan et al., 2017), and incorporated them into their training objective function to enhance the robustness of the concepts. They have also demonstrated the numerical improvement in stability.

**Strengths:**

1-I really liked the paper. The authors address a very important question of model interpretability, which pertains to the robustness of explanations, in this case, regarding concepts.

2-The authors not only discussed conditions for ensuring the robustness of concepts but also incorporated them into a well-defined objective function.

3-I also appreciated the numerical comparison of stability in the experiments, which many papers only illustrate with a few examples.

4- Finally, in my opinion, the authors conducted significant experiments to support their claims.

**Weaknesses:**

The quality of the explanations for some parts of the paper could be enhanced.

1- It took me a significant amount of time to follow the flow of data through the network. Perhaps adding concise explanations detailing how input images traverse the network during inference would make the explanation more accessible.

2- Likewise, when explaining the framework, the authors could introduce all the key terms at the beginning of the section so that readers do not need to continuously reference earlier sections to grasp the meanings of these terms.

**Questions:**

1- Can the authors explain the choice of architecture for the text and image encoder and its impact on the model's performance, particularly in terms of accuracy?

2- Also, which image model is used in "Standard (No interpretability)" in table 1.

---

> ### Author Response · Authors · 2023-11-20
> **Response to Ak8W**
>
> Dear Reviewer **Ak8W**,
>
> We would like to express our sincere gratitude for taking the time to review our paper and providing valuable feedback. We have carefully considered your comments and have made revisions to address each of the issues raised. Please find our responses to your specific points below:
>
> **Response to W1:**
>
> In the revised version, we have added more details to all figures (including Figure 2) in our paper.
>
> **Response to W2:**
>
> In the revised version, we have added a notation Table 4 in Appendix (page 13). We will move it to the main context if additional pages are allowed.
>
> **Response to Q1:**
>
> Our text encoder and image encoder make up CLIP. The text encoder of CLIP is used to convert natural language text into vector representations. It is based on the Transformer model architecture, which learns to embed text into a high-dimensional vector space, where text with similar semantics are closer in the vector space. The text encoder transforms input text sequences into fixed-dimensional vector representations, capturing the semantic and contextual information of the text. The image encoder of CLIP is used to convert images into vector representations. It employs a convolutional neural network (CNN) as the backbone model to extract image feature representations. These feature representations are mapped to a fixed-dimensional vector space for comparison with the vectors from the text encoder.
> By using the pre-trained CLIP text encoder and image encoder, our model is capable of understanding both images and text (concepts). By encoding images and text into the same vector space, CLIP quantifies the semantic similarity between images and text, enabling joint encoding and understanding across modalities. CLIP leverages large-scale pre-training datasets, allowing it to learn rich semantic representations. The advantage of pre-training is the ability to learn general-purpose feature representations from large-scale data. It is important to emphasize that we did not train the image encoder and text encoder ourselves.
> We use the encoder to obtain the image-concept matrix to complete the concept-neuron (feature) mapping. In fact, this is the process that affects the interpretability of the model, and whether the choice of encoder affects accuracy is something we have not discussed. Because we couldn't find another image-text encoder model that was pre-trained on large-scale datasets for comparison.
>
> **Response to Q2:**
>
> The standard model represents an image classification model, which extracts image features through the same backbone as our FVLC model and then connects a fully connected layer to complete the image classification task. In fact, this is a common practice in the field of CBM. After adding the bottleneck layer, the accuracy of the image classification task will be affected due to the intervention of the concept task.  We have added all baseline settings in the revised version (Section D.3 in Appendix - page 16).

---

### Author Response · Authors · 2023-11-20
**To everyone**

In the revised version

**(1)** We improved all figures and tables to make them more readable.

**(2)** We have added a notation Table 3 in Appendix (page 13).

**(3)** We added more details on the baseline methods (Section D in Appendix).

**(4)** We also add more intuition for our definition of FVLC, and more details on the definition of FVLC in Section A of Appendix.

**(5)** We have added some references to the related work section to make it more complete.

**(6)** We did some experiments on the HAM10000 dataset (medical dataset) to improve the reliability of our model, as detailed in Appendix Section H, Table 12, and Figure 9.

**(7)** We have enlarged the subfigures in Figure 3, see Figure 5-8 for details.

**(8)** We have polished the definition of FCLV, and the optimization objective function, see Page 5, 6 and Appendix Section B for details.

**(9)** In our response to Reviewer UywT, we add some ablation studies on the performance of adversarial attacks. Also, we add some results on the variance.

---

### Meta-Review · Area_Chair_Jpd6 · 2023-12-11

**Metareview:**

This paper contributes an in-depth analysis of faithful concepts in the state-of-the-art Label-free concept bottleneck model (LCBM), where LCBM is a recent framework that has addressed a critical challenge in CBM field by building CBMs without the need of concept labels.
The reviewers generally agree with the motivation and value the importance of this paper to ensure faithful vision-language concepts (FVLC). The reviewers also find the numerical experiments are extensive and convincing and support the claims.

There was a mixed rating on the paper clarity (some reviewers think it's easy to follow while some thinks it's poorly written), and it has been addressed by the revised manuscript in the rebuttal.

Based on the above reason, an acceptance (poster) is recommended.

**Justification For Why Not Higher Score:**

This paper tackles an important problem by building upon previous work, which has the contribution level between a ICLR poster or spotlight.

**Justification For Why Not Lower Score:**

The technical contribution of this paper is useful to the interpretable ML community.

---

### Decision · Program_Chairs · 2024-01-16

Accept (poster)